Resource


# SimuCell3D: three-dimensional simulation of tissue mechanics with cell polarization



Steve Runser[1,2], Roman Vetter[1,2] & Dagmar Iber[1,2]

The three-dimensional (3D) organization of cells determines tissue function and integrity, and changes markedly in development and disease. Cell-based simulations have long been used to define the underlying mechanical principles. However, high computational costs have so far limited simulations to either simplified cell geometries or small tissue patches. Here, we present SimuCell3D, an efficient open-source program to simulate large tissues in three dimensions with subcellular resolution, growth, proliferation, extracellular matrix, fluid cavities, nuclei and non-uniform mechanical properties, as found in polarized epithelia. Spheroids, vesicles, sheets, tubes and other tissue geometries can readily be imported from microscopy images and simulated to infer biomechanical parameters. Doing so, we show that 3D cell shapes in layered and pseudostratified epithelia are largely governed by a competition between surface tension and intercellular adhesion. SimuCell3D enables the large-scale in silico study of 3D tissue organization in development and disease at a great level of detail.

The acquisition and maintenance of proper morphology are crucial for the normal physiological functioning of a biological tissue. Their disruptions are associated with a range of pathological conditions, including cancer and birth defects. The shape of tissues is determined by the dynamic positioning of their constituent cells, which can collectively deform or migrate to induce macroscopic changes in the tissue morphologies[1,2]. These cellular behaviors are regulated by the mechanical properties of both cells and extracellular matrix (ECM)[3], along with the distribution of stresses within tissues[4]. Therefore, understanding how tissues acquire and maintain their shapes requires a deep comprehension of the interplay between the stress distribution within them and the mechanical properties of their cells and ECM.

Various experimental methods have been developed to contribute to this understanding[5–7]—for example, micropipette aspiration[8], atomic force microscopy[9], optical stretcher[10] and laser ablation[11]. Nonetheless, these experimental techniques are generally limited to the rare tissues directly accessible to probing, or to small tissue portions. In addition, even when all the factors influencing a tissue morphology have been experimentally identified, their synergy might remain unclear.

Recent advances in the fields of fluorescent microscopy, image processing and computation power now allow us to complement these direct measurements with in silico models, and thus to gain a more global understanding of the cellular dynamics underlying tissue morphogenesis and homeostasis[12–17]. Among these numerical methods, cell-based models have become widely used in the fields of developmental and cancer biology due to their high spatiotemporal resolution and accurate predictions. Cell-based models recreate virtual versions of tissues by representing cells as individual agents with their own mechanical properties and behavior. These models offer an in silico environment where the stress distribution and the mechanical properties of cells can be modulated to study their impact on tissue morphology and function. Cell-based models can thus predict the tissue shape arising from experimentally measured cell properties or, conversely, in conjunction with parameter estimation methods, they can allow us to infer the cell properties that led to an imaged tissue morphology. The high level of spatiotemporal details of cell-based models, however, entails a substantial computational cost, which forces a trade-off between the number of cells they can simulate and the spatial resolution of their representation[18]. For this reason, cell-based models with varying levels of resolution have been developed to address different types of biological problem. For instance, center-based models are a class of cell-based models that represent cells as simple spheres, making

[1]Department of Biosystems Science and Engineering (D-BSSE), ETH Zürich, Basel, Switzerland. [2]Swiss Institute of Bioinformatics (SIB), Basel, Switzerland. e-mail: dagmar.iber@bsse.ethz.ch

**Table 1 | Comparison of SimuCell3D with existing 3D DCM models**

| Program name | Pub. year | No. of cells after 1d of computation | Adjustable spatial resolution | Automatic mesh remodeling | Cell divisions | Cell polarization | Nuclei | ECM | Lumen | OS | Lic. | Ref. |
|---|---|---|---|---|---|---|---|---|---|---|---|---|
| SimuCell3D | 2024 | 125,000 | ✓ | ✓ | ✓ | ✓ | ✓ | ✓ | ✓ | ✓ | BSD-3 | — |
| | 2023 | unknown | ✓ | ✓ | ✓ | ✗ | ✗ | ✗ | ✗ | ✗ | — | 51 |
| | 2023 | unknown | ✓ | ✓ | ✓ | ✓ | ✗ | ✗ | ✗ | ✗ | — | 52 |
| IAS | 2022 | 4 | ✓ | ✓ | ✓ | ✗ | ✗ | ✗ | ✗ | ✓ | CC | 47 |
| | 2020 | <1,000 | ✓ | ✗ | ✓ | ✓ | ✓ | ✗ | ✗ | ✗ | — | 45 |
| CellSim3D | 2018 | 75,000 | ✗ | ✗ | ✓ | ✗ | ✗ | ✗ | ✗ | ✓ | GPLv2 | 50 |
| | 2014 | starting number | ✓ | ✓ | ✗ | ✗ | ✗ | ✗ | ✗ | ✓ | BSD-2 | 44 |
| | 2013 | starting number | ✓ | ✓ | ✗ | ✗ | ✗ | ✗ | ✗ | ✗ | — | 53 |
| The Surface Evolver | 1992 | starting number | ✓ | ✗ | ✗ | ✗ | ✗ | ✗ | ✗ | ✓ | — | 49 |

Cell numbers are approximate. Pub., publication. Lic., license. OS, open source. Ref., reference. CC, Creative Commons. GPL, GNU General Public License. BSD, Berkeley Software Distribution.

them suitable for phenomena where the abundance of cells is more crucial than their shape. These models have been used to gain deeper understanding of a wide range of phenomena, including, for instance, the development of tumors[19] or the inflammation of tissues[20].

Vertex models are another class of cell-based model that have been developed to study tissues in which cell shapes can be approximated by polygons in two dimensions[21–25] or polyhedra in three dimensions[26]. This simplification allows them to represent each cell with only a few points, enabling them to simulate large tissues. Vertex models have been employed to study a wide range of phenomena, including the transition between solid-like and fluid-like tissue states[27], as well as various morphogenetic processes such as the polarization of early embryos[28], the formation of branched structures[29] and the biased elongation of tissues[30,31]. However, their simplistic representation of tissues comes with the drawback that they cannot adequately represent cells with complex shapes. Furthermore, the highly restricted topology permissible for the mesh in vertex models substantially complicates the simulation of phenomena such as cell extrusion or tissue fusion. The mechanisms underlying these developmental events are among the fundamental open problems in morphogenesis.

To address the limitations of vertex models, a family of cell-based models sometimes referred to as deformable cell models (DCMs) has been developed. These models provide a more geometrically realistic representation of tissues by discretizing each cell membrane separately into a closed loop of connected points in two dimensions[32–43] or a closed triangulated surface in three dimensions[44–53]. The complex shapes that cells can adopt in DCMs make these models particularly suited to simulate phenomena such as the development of early embryos[54] or the cellular movements during wound healing[45]. However, the accuracy offered by these models comes at a staggering computational cost. To mitigate this computational cost, one 3D DCM implementation named CellSim3D[50] constrains the cell geometries to spheroidal shapes. This approach is however not suited for the study of tissues with complex (non-polyhedral) cell shapes. The remaining 3D DCMs preserve a high geometrical resolution of the cell membranes but are limited by their computational efficiency. At best, they can simulate the growth of a tissue from one to a thousand cells in a week of computation time[45], precluding their use for large-scale computational studies. Additionally, the numerical stability of these models may be compromised when the simulated cells undergo large deformations. We review the features of available 3D DCMs in Table 1.

Here we present SimuCell3D, an efficient open-source DCM in three dimensions. Thanks to its efficient design, SimuCell3D can simulate tissues composed of dozens of thousands of cells with high spatial resolution. SimuCell3D overcomes the classical trade-off that has so far constrained cell-based models between their resolutions and the number of cells they can simulate. In addition, our program natively allows us to represent intra- and extracellular entities such as nuclei, lumens, ECM and non-uniform mechanical cell membrane properties, as found in polarized cells (Fig. 1a). By combining speed and versatility, SimuCell3D can simulate processes that had not been amenable to existing numerical methods.

# Results

## Biophysical model

SimuCell3D aims to simulate the morphodynamics of cellular tissues at a high spatial resolution with full account of complex cell shapes. The shapes and motion of the simulated cells are not constrained by the model representation, and their mechanical properties are based on the physical principles governing the dynamics of their biological counterparts. This unconstrained representation of the cells is achieved by modeling their surfaces with disjoint closed triangulated surfaces (Fig. 1b). The spatial resolution of these surfaces can be tuned by adjusting the size of their triangles. To ensure that the simulations are initialized at the desired resolution, a custom triangulation algorithm has been incorporated into SimuCell3D (Supplementary Fig. 1), allowing the use of geometries obtained from microscopy images as the starting point of the simulations. A local remeshing algorithm (Supplementary Fig. 2) preserves the mesh resolution and quality even under large cell deformations. Apart from viscous damping as well as repulsive and adhesive cell–cell contacts, the biomechanical state of each cell membrane is defined by the following energy potential (Fig. 1c):

$$U = KV\left(\ln\frac{V}{V_0} - 1\right) + \frac{k_a}{2}\left(\frac{A}{A_0} - 1\right)^2 + \int_{\partial\Omega}\left(\gamma + \frac{k_b}{2}(2H)^2\right)\mathrm{d}S. \quad (1)$$

The first term is the energy associated with a net internal pressure, $p = \mathrm{d}W/\mathrm{d}V = -K\ln(V/V_0)$, which arises from the volumetric strain of the cell cytoplasm, modeled as a slightly compressible fluid. $W$ denotes work, $V$ and $V_0$ are the current and target cell volumes and $K$ the cytoplasmic bulk modulus. Shrinkage or growth of cells can be achieved by evolving their target volumes in time. The second energy term allows each cell to actively regulate its membrane area $A$ by penalizing deviations from a target value $A_0$ with an effective isotropic membrane elasticity parameter $k_a$. The first term in the surface integral, which runs over the cell surface $\partial\Omega$, models the tension generated by the cell actomyosin cortex. $\gamma$ is the isotropic cortical tension, analogous to the surface tension of fluid interfaces. The second integrand models the resistance of the cell cortex to bending[55], with $H$ denoting the local mean curvature of the cell membrane and $k_b$ its bending rigidity. $\gamma$ and $k_b$ are field parameters that can vary along the cell surface according to cell polarity (Fig. 1b).


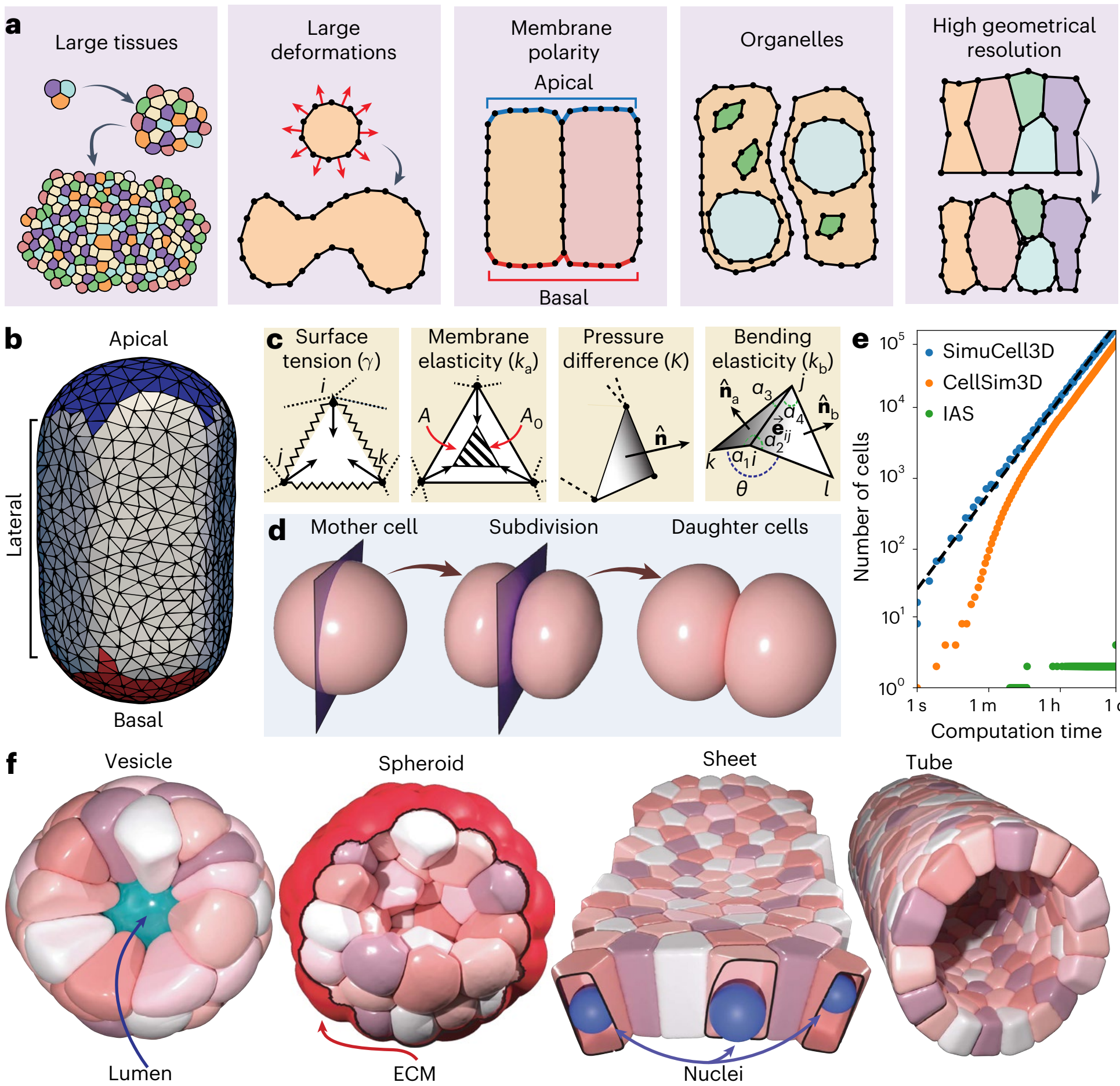


**Fig. 1 | Representation of cellular tissues in SimuCell3D. a**, Schematics of the features that cell-based models must possess to be applicable to a broad range of morphogenetic problems. **b**, Representation of cell membranes as closed triangulated surfaces with non-uniform material properties on the basis of cell polarity (colors). **c**, Summary of different forces acting on the triangulated cell membranes. **d**, Illustration of cell division perpendicular to a division plane (purple). **e**, Computational efficiency of different 3D cell-based models in an exponential-growth scenario. The dashed black line is a fitted power law $T = aN_c^{\alpha}$ with coefficients $a = 0.013$ s and $\alpha = 1.33$. **f**, Illustration of different tissue topologies and intra- or extracellular features that can be simulated with SimuCell3D. All surfaces can be non-convex.

SimuCell3D offers two distinct contact models to simulate intercellular interactions. The first model mediates interactions through local elastic contact forces, taking into account cell–cell adhesion and volumetric exclusion (Methods, equation (2), and Supplementary Fig. 3). Its two constitutive parameters, the adhesion strength $\omega$ and the repulsion strength $\xi$, are field quantities that can vary among cells or membrane regions. This contact model is somewhat dependent on mesh resolution (Supplementary Fig. 4), just as adhesion in biology will depend on the adhesion protein density. The second contact model mechanically couples the nodes of adjacent cells and directly transfers forces generated on one cell surface to that of the neighboring cell. We validated this second model by reproducing the Young–Dupré relationship in cell doublets and triplets (Supplementary Fig. 5a,c). The resulting contact mechanics are independent of mesh resolution (Supplementary Fig. 5a). All parameters related to intercellular interactions are summarized in Table 2.

SimuCell3D can simulate entities such as nuclei, lumens and ECMs by also representing them with closed triangulated surfaces similarly to the cell membranes. To model cell death, cells can be removed from the tissue if their volume drops below a minimum threshold $V_{min}$. Conversely, if cell volumes exceed the maximal value $V_{max}$, they undergo cytokinesis (Fig. 1d). The division plane can be randomly oriented or perpendicular to the longest cell axis (Hertwig's rule). A cell division only takes a few microseconds of computation time, allowing the simulation of tissues with high cell division rates. To demonstrate the computational efficiency and stability of our program, we simulated the exponential growth of a tissue in an out-of-equilibrium regime with the growth rate pushed to the limit, starting from a single cell (Fig. 1e and Supplementary Video 1). The cells in this test are simulated without nuclei. Only one day of computation time is required to grow the tissue to 125,000 cells on an Intel Xeon W-2245 CPU (eight cores, 3.9 GHz) using 16 threads, for cells that possess 121 nodes and 238 triangular faces on average. The total time complexity of such a simulation is $\mathcal{O}(N_c^{4/3})$, where $N_c$ is the number of cells in the tissue, which is equivalent to the scaling observed in two-dimensional simulations[43]. Under similar settings, we tested the performance of CellSim3D[50] and Interacting Active Surfaces (IAS)[47], two other cell-based 3D models offering low and high spatial resolution, respectively. CellSim3D generated a tissue of 75,000 cells in a day of computation time while IAS produced a tissue of 4 cells in the same amount of time. CellSim3D achieves performance comparable to that of our program by constraining the cell geometries to simple spherical shapes with a fixed number of nodes. SimuCell3D thus offers the performance of low-resolution models such as CellSim3D while possessing the flexibility and accuracy of high-resolution models such as IAS. SimuCell3D is parallelized with OpenMP. The parallelization efficiency follows Amdahl's law (Supplementary Fig. 6). To showcase the versatility of our program, we simulated various tissue topologies such as a vesicle, a bulk spheroid, a sheet and a tube,

**Table 2 | Model parameters**

| Symbol | Default dynamic | Default overdamped | Measured | Unit | Dimension | Description | Ref. |
|---|---|---|---|---|---|---|---|
| **Cell volume parameters** | | | | | | | |
| $\rho$ | 1,000 | 1,000 | 1,045–1,099 | $kg\,m^{-3}$ | $M/L^3$ | Mass density | 77 |
| $K$ | 2,500 | 2,500 | 2,250 | Pa | $M/LT^2$ | Bulk modulus | 78 |
| $p_{max}$ | 2,500 | 2,500 | 300–2,200 | Pa | $M/LT^2$ | Max. internal net pressure | 79–81 |
| $V_{min}$ | $3.7\times10^{-16}$ | $3.7\times10^{-16}$ | $2.5–3.7\times10^{-16}$ | $m^3$ | $L^3$ | Min. volume (apoptosis) | 64,82 |
| $V_{max}$ | $1.4\times10^{-15}$ | $1.4\times10^{-15}$ | $0.9–1.3\times10^{-15}$ | $m^3$ | $L^3$ | Max. volume (cell division) | 64,82 |
| $g$ | $10^{-11}$ | $10^{-11}$ | $0.1–1.8\times10^{-20}$ | $m^3s^{-1}$ | $L^3/T$ | Volumetric growth rate | 83,84 |
| **Cell surface parameters** | | | | | | | |
| $\gamma$ | 0.001 | 0.001 | 0.0005–0.0025 | $N\,m^{-1}$ | $M/T^2$ | Surface tension | 81,85,86 |
| $k_b$ | $2\times10^{-18}$ | $2\times10^{-18}$ | $1–2\times10^{-18}$ | J | $ML^2/T^2$ | Bending stiffness | 87 |
| $k_a$ | $10^{-15}$ | $10^{-15}$ | — | J | $ML^2/T^2$ | Area elasticity modulus | — |
| $Q_0$ | 250 | 250 | 300 | — | — | Target isoparametric ratio | 64 |
| $\xi$ | $10^9$ | $10^9$ | — | $Pa\,m^{-1}$ | $M/L^2T^2$ | Repulsion strength | — |
| $\omega$ | $10^9$ | $10^9$ | — | $Pa\,m^{-1}$ | $M/L^2T^2$ | Adhesion strength | — |
| $H_{max}$ | $5\times10^6$ | $5\times10^6$ | — | $m^{-1}$ | 1/L | Max. coupling curvature | — |
| **Numerical parameters** | | | | | | | |
| $l_{min}$ | $2\times10^{-7}$ | $2\times10^{-7}$ | — | m | L | Minimum edge length | — |
| c | $2\times10^{-7}$ | $2\times10^{-7}$ | — | m | L | Contact cutoff distance | — |
| $\zeta$ | $2.5\times10^{-10}$ | $3\times10^{-9}$ | — | $kg\,s^{-1}$ | M/T | Viscous damping coefficient | — |
| $\Delta t$ | $10^{-7}$ | $10^{-7}$ | — | s | T | Time step | — |

Default parameter values are given for the two types of equation of motion implemented in SimuCell3D (dynamic versus overdamped). In the parameter dimension, M represents mass, L length and T time. Default values produce a typical tissue growth scenario.

alongside several intra- or extracellular built-in features such as lumens, nuclei and ECM (Fig. 1f and Supplementary Video 2).

### Cell membrane polarization

Cells form regions with distinct biochemical and mechanical properties along their cytoplasmic membranes. Correct establishment of this cell polarity is crucial to numerous developmental processes[56]. Its impairment has also been linked to the onset of tumor formation[57]. SimuCell3D takes cell polarity into account by allowing the triangular faces to be of different types with distinct mechanical parameters $\gamma$, $k_b$, $\omega$ and $\xi$. Two mechanisms are implemented to automatically identify different regions on the cell surfaces. In the first, lateral sides are inferred from the face contact information, leaving regions that are not in contact as either apical or basal. The second, more robust and versatile, algorithm is based on a spatial partitioning of the simulation domain into voxels representing one of four possible regions: cell boundaries, luminal, cytoplasmic and external (Fig. 2a–c). Voxels containing mesh nodes are marked as boundary voxels. The remaining unmarked voxels are clustered with the Hoshen–Kopelman algorithm[58]. The different voxel clusters thus created are then labeled as cytoplasmic, luminal or external on the basis of their positions in the discretized simulation space. Then, each surface triangle probes its environment by casting a ray in the direction of its outward normal to detect which type of region it faces (Fig. 2d). The type of voxel the ray first passes through determines whether the mesh triangle is lateral (facing another cell), apical (facing an enclosed volume such as a lumen) or basal (facing the surrounding medium or ECM). Iteration over all mesh triangles thus tags the entire surface (Fig. 2e). We demonstrate the capabilities of this approach by reproducing in silico a monolayer prostate organoid whose cells exhibit apicobasal polarity (Fig. 2f). The cell surfaces were extracted from 3D microscopy images with Cellpose[59] (Fig. 2g). SimuCell3D then reproduced the organoid with correct tissue polarity (Fig. 2h) without requiring any input on tissue orientation or topology by the user.

### Application 1. Transition from monolayer to multilayer tissue

We now demonstrate how SimuCell3D can be used to gain insight into the cellular dynamics of biological tissues. As a first showcase, we investigate the relationship between biomechanical cell parameters and the internal structure of a tissue as a mono- or multilayer. Such a difference in cellular organization is particularly striking between different types of epithelial tissue[60]. Strong evidence suggests that this variability is the result of an interplay between intracellular surface tension and intercellular adhesion[61,62]. In a tug of war with cortical tension, in which the actomyosin cortex tends to minimize the cell surface area, adhesion molecules between adjacent cells tend to increase it. We investigated this competition by numerically exploring the parameter space spanned by adhesion strength and surface tension. The simulations were initialized with a spherical monolayer vesicle consisting of 432 columnar cells generated from a Voronoi tessellation of the sphere (Fig. 3a). All cells were initially in contact on their apical sides with a luminal region and on their basal sides with an ECM encasing the tissue. Note that no ECM located at the apical side of the cells nor any adhesion belt was considered in these simulations. The cells were grown at a uniform volumetric rate without division until they had doubled in size, while the luminal target volume was preserved. Despite cellular rearrangements caused by growth, we observed the maintenance of the monolayer structure in simulations with low surface tension (Fig. 3b). Strong cortical tension, on the other hand, leads to stratification (Fig. 3c). We quantified the resulting number of cell layers by converting the tissue into a graph representing cell connectivity and computing the shortest path percolating from the lumen to the ECM (Fig. 3d). Parameter values were non-dimensionalized with $l = \langle V(t=0)\rangle^{1/3}$ as a characteristic length scale, and $K$ as a characteristic energy density. Our exploration of the parameter space revealed that, under the prescribed conditions, the layering of the tissue is essentially regulated by the tension of the actomyosin cortex alone (Fig. 3e). An increase in the normalized surface tension $\tilde{\gamma} = \gamma/Kl$ from 0.02 to 0.10

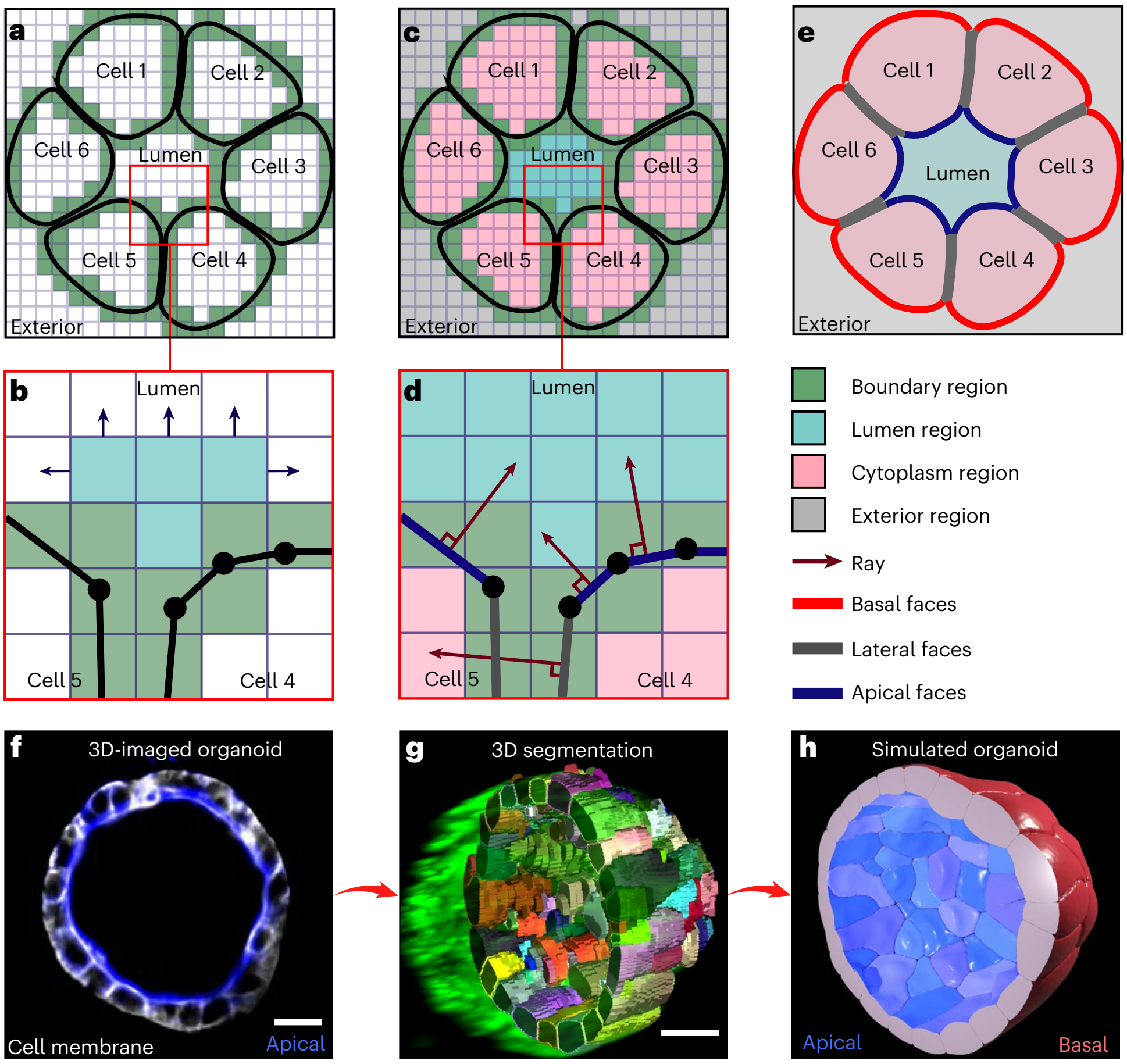


**Fig. 2 | Automatic cell surface polarization algorithm.** For visual clarity, the process is shown in two dimensions. **a**, The simulation space is discretized with uniformly sized cubic voxels. All voxels that intersect with the cell surfaces are marked as boundary voxels. **b**, All remaining voxels are clustered with the Hoshen--Kopelman algorithm. **c**, Voxel clusters are tagged as lumen, cytoplasm or exterior. **d**, A ray is cast from the center of each face in the outward normal direction. The first voxel (other than a boundary) that it intersects with indicates with which region the face is in contact. **e**, Facial types are assigned on the basis of the regions with which they are in contact. **f**, Cross-sectional light-sheet microscopy image of a mouse prostate organoid. Cell polarity is visualized by Ezrin staining (blue, apical side). **g**, 3D cell segmentation of the organoid. **h**, Simulated organoid with automatically polarized epithelial surfaces (blue, red). The different membrane regions can possess different surface tensions, bending rigidities, adhesion strengths and repulsion strengths. Scale bars, 15 µm.

was sufficient to break the monolayer arrangement and force the tissue into a stratified structure. Conversely, an increase by two orders of magnitude in the normalized adhesion strength $\tilde{\omega} = \omega l/K$ between the cells did not disrupt the monolayer integrity. As the cells lose their apicobasal connectivity at stronger surface tension, they adopt a more spherical shape that minimizes their surface area, as measured by their sphericity $\Psi = \pi^{1/3}(6V)^{2/3}/A$ (Fig. 3f). These simulations highlight the potential of SimuCell3D to quantitatively address open questions in tissue development and cancer progression, the latter being linked to a loss of structural tissue integrity on the cellular level[63].

## Application 2. Formation and maintenance of pseudostratification in epithelia

Pseudostratified epithelia are single-layered epithelia that are easily mistaken as stratified when analyzed in two-dimensional sections because of the dispersion of their nuclei along the apical–basal axis[64]. Their ubiquity across different species during development[65] suggests that the pseudostratified structure can confer an advantage over simpler cellular arrangements, possibly linked to patterning precision[66]. How this structure is acquired and maintained under growth and morphogenetic deformation is still largely unknown. In this second case study, we demonstrate how SimuCell3D may be used to gain mechanistic insight into the elusive pseudostratification process. We initialized simulations with a patch of 70 cells segmented from light-sheet microscopy images of the developing pseudostratified mouse lung epithelium[64] (Fig. 4a). Among these 70 cells, the 21 interior cells were allowed to move freely while the rest on the periphery of the patch acted as static boundaries. The simulated cells all contained a nucleus (Fig. 4b, blue) and neither grew nor divided during the simulations, but deformed to minimize their potential energy, until static equilibrium was reached. We again examined the interplay between cell surface tension ($\tilde{\gamma}_c = \gamma_c/l_c K_c$, subscript 'c' for cell) and adhesion strength ($\tilde{\omega}_c = \omega_c l_c/K_c$) (Fig. 4c,d). The normalized surface tension of the nuclei ($\tilde{\gamma}_n = \gamma_n/l_n K_n$, subscript 'n' for nucleus) was kept constant at 0.24 in these simulations, and they were non-adhesive ($\omega_n = 0$). In the explored region of the parameter space, we observed two unphysiological morphological cell regimes (I and II) with a continuous transition in between, along which an intermediate physiological range can be identified (Fig. 4c). In regime I, the cell shape is dominated by the effect of surface tension. Some of the cells segregated in response to the strong surface-area minimization tendency (Fig. 4c, left), facilitated by weak lateral adhesion. Cells in this regime reduced their lateral cell–cell contact area fraction $\phi$ (Fig. 4d) and also possessed fewer neighbors, as measured by their coordination number $z$ (Fig. 4e). In regime II, the effect of adhesion dominates over surface tension, allowing neighboring cells to maximize their mutual contact areas (Fig. 4c, right; Fig. 4d) as well as their coordination number (Fig. 4e). In between these extremes, a balance between adhesion strength and cortical tension

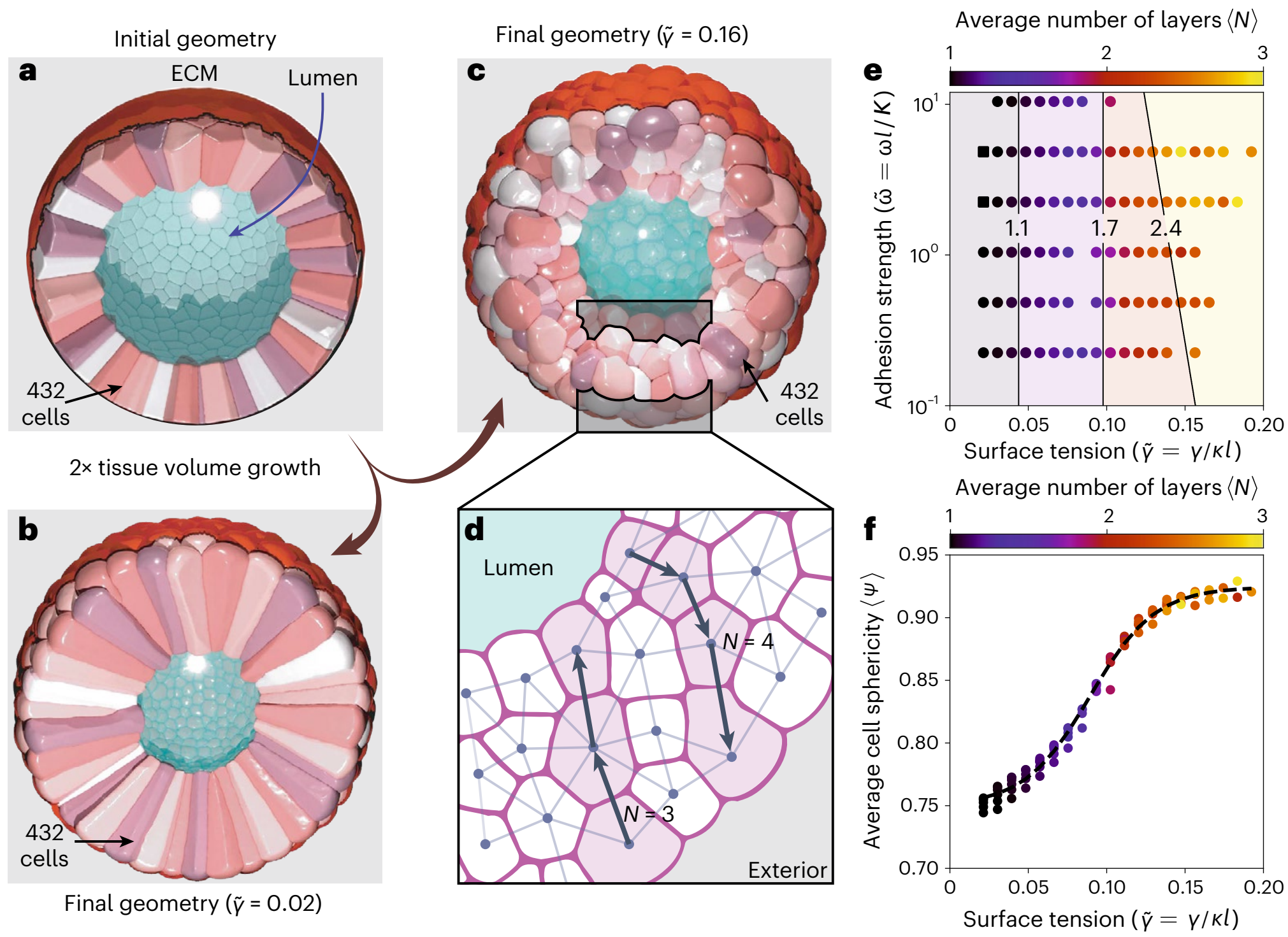


**Fig. 3 | Impact of biomechanical cell properties on tissue structure.** **a**, Initial tissue geometry: a hollow spherical vesicle made up of 432 columnar epithelial cells. The central luminal region (turquoise) was represented by a non-growing volume. On its basal side, the tissue was encased by a surface representing the ECM (red). **b**, Final monolayer conformation after epithelial volume doubling with weak cortical tension. **c**, Final multilayer conformation at strong cortical tension. **d**, Schematic cross-section through the epithelium with a cell connectivity graph on which the layer number *N* was determined. **e**, Impact of cell surface tension and adhesion strength on the average number of cell layers. Isolines represent support vector machine discriminants. Each data point corresponds to the final state of one numerical simulation. Simulations in which the whole tissue remained a monolayer ($N \equiv 1$) are shown as squares. **f**, Effect of cortical tension on cell shape, as measured by sphericity, and on the number of layers (colors). The dashed black curve is a fitted logistic function.

yields physiological cell shapes corresponding to those imaged (Fig. 4c, middle). This morphological similarity can be exploited to infer the mechanical properties of in vivo pseudostratified cells (Fig. 4d,e). Moreover, besides informing on the mechanical state of cells, SimuCell3D unveiled in this second case study the possibility that pseudostratified tissues could be formed from cells with identical mechanical properties.

Subsequently, we used SimuCell3D to investigate the effect of mechanical properties of the nuclei on the pseudostratified cell organization (Fig. 4f,g). In these simulations, the cell surface tension $\tilde{\gamma}_c = 0.01$ and adhesion strength $\tilde{\omega}_c = 0.97$ were fixed. By varying the nuclear surface tension $\tilde{\gamma}_n$, we were able to create nuclei rigid enough to deform the cell membranes (Fig. 4f). Cell deformation was measured by comparing the equilibrium cell shape in the presence of a nucleus versus that in its absence, quantified by the intersection over union: $\chi = 1 - \mathrm{IoU}(\Omega \text{ with nucleus}, \Omega \text{ without nucleus})$. We observe an increase of the average cell deformation $\langle\chi\rangle$ with the nucleus surface tension $\tilde{\gamma}_n$ until the nuclei obtain spherical shapes at $\tilde{\gamma}_n \approx 0.35$. It then saturates at $\langle\chi\rangle \approx 0.13$ as nuclear tension increases further. The average sphericity of the nuclei has been measured in the segmented geometries at 0.89, suggesting a low cortical stiffness of the nuclear envelopes relative to the cytoplasmic membranes.

SimuCell3D also allows us to directly modulate the shapes of nuclei or cells by concurrently varying their target isoperimetric ratios $Q_{0,n} = A_{0,n}^3/V_{0,n}^2$, and area elasticity moduli $k_{a,n}$ (Fig. 4g). Simultaneously increasing $Q_{0,n}$ and $k_{a,n}$ drives the equilibrium shapes of nuclei away from a sphere. Conversely, nuclei with small values of $Q_{0,n}$ and $k_{a,n}$ possess more spherical shapes. The ability to thus change the stiffness or shape of the nuclei opens up opportunities to study the dynamics of interkinetic nuclear migration[67].

## Discussion

SimuCell3D now permits the in-depth in silico investigation of the mechanical properties and behavior of cells to understand the mechanisms that regulate tissue homeostasis and morphogenesis. While the current simulations were carried out with linear mechanical models, nonlinear material behavior (viscoelasticity, hyperelasticity) could readily be implemented to study its effect on morphogenesis. Moreover, besides nuclei, organelles and endocytosis could be easily represented. As such, processes such as interkinetic nuclear migration in pseudostratified epithelia could be simulated at unprecedented resolution to address open questions regarding the driving forces.

As we showed, SimuCell3D can be used to predict the global tissue morphologies that emerge from individual mechanical cell properties. Specifically, when the morphological features of the tissues are known, SimuCell3D can be used to infer the region of the mechanical parameter space in which the cells are located. Our exploration of the cellular parameter space in this study was mainly limited to the subspace spanned by cell cortical tension and adhesion strength. This subspace is insufficient to reproduce the wealth of morphogenetic events observed in vivo. In other contexts, exploration of higher-dimensional parameter spaces will undoubtedly be necessary. In these circumstances, SimuCell3D can be coupled with gradient-free parameter estimation techniques to accurately infer the cell properties that lead to the measured morphological tissue features.

SimuCell3D is readily extendable to accommodate more features in the future. Relevant possible extensions include subcellular components such as adhesion belts, frictional forces, (which play an important role in the morphogenesis of some tissues[68]) as implemented in pre-existing models[45,52], tension fluctuations[69] and reaction–diffusion models to couple the biomechanical tissue model with chemical


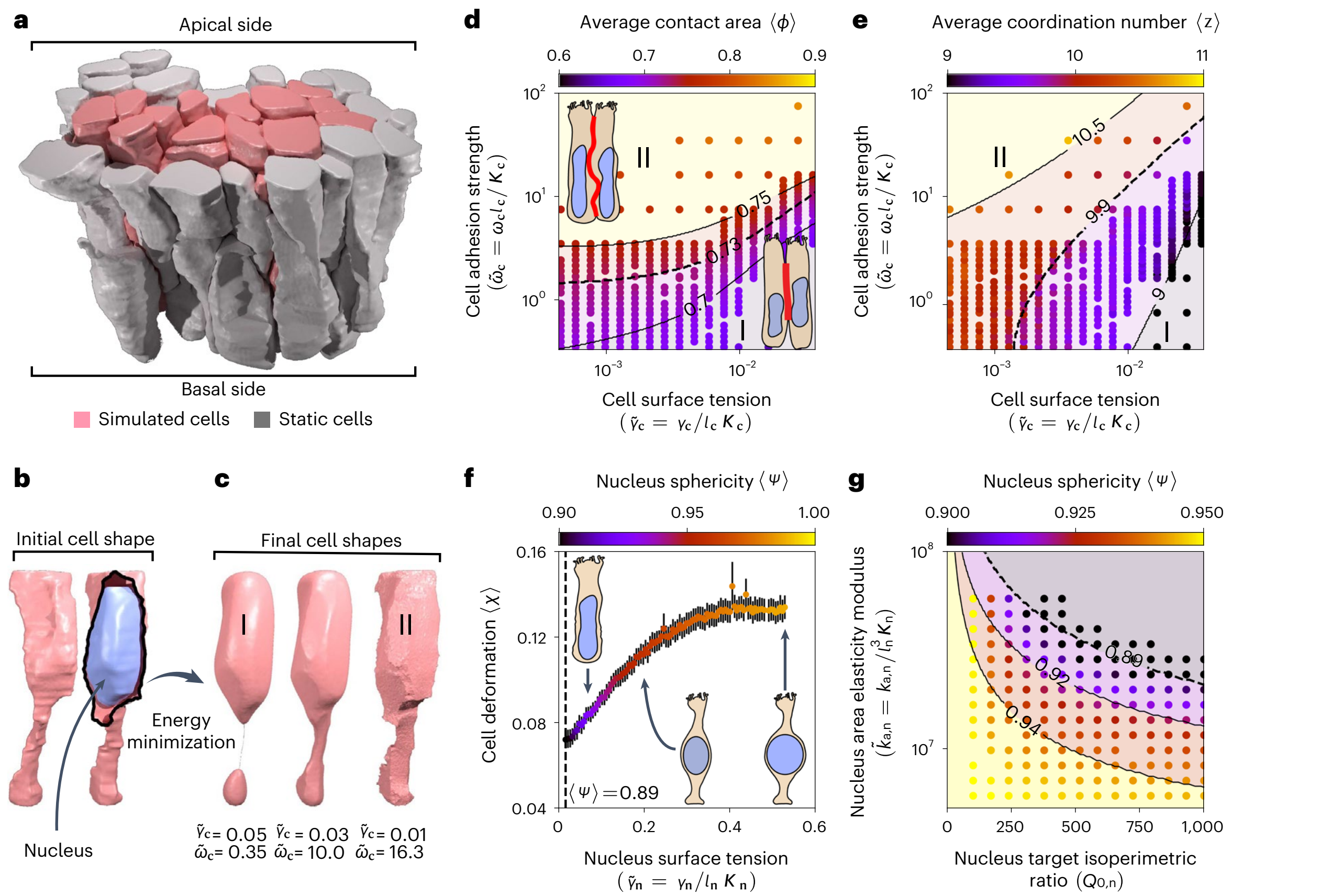


**Fig. 4 | Simulation of 3D cell organization pseudostratified epithelia. a**, Initial geometry imported into SimuCell3D: a patch of 70 cells from the developing pseudostratified mouse lung epithelium, segmented and triangulated from light-sheet imaging[64]. 49 gray cells act as rigid boundaries for the 21 simulated pink cells. **b**, All cells contain a nucleus (blue). **c**, On the basis of the cell surface tension and adhesion strength, the cells adopt different shapes (regimes I and II). **d**, Mean fraction of cell surface area in contact with other cells (red line in illustration), as a function of the adhesion strength and cortical tension. **e**, Mean cell coordination number in the same parameter space. **f**, Mean cell deformation as a function of the nuclear surface tension. Error bars represent the s.e.m., $n = 21$ nuclei. **g**, Average nuclear sphericity as a function of the nucleus area elasticity modulus and target isoperimetric ratio. Isolines in **d**,**e**,**g** represent support vector machine discriminants. Dashed curves in **d**–**g** represent the physiological values measured from the segmented geometries. Each data point corresponds to the equilibrium state obtained from one numerical simulation.

signaling. In this way, chemical and mechanical symmetry-breaking mechanisms could be combined and their effects could be simulated at cellular resolution. Finally, the cell-based simulations could be combined with continuum models to simulate the behavior of larger tissues at varying resolution, and to derive adequate material models for the continuum description from cell-based simulations.

# Methods

## Mesh operations

**Local mesh adaptation.** SimuCell3D geometrically represents cells by closed triangulated surfaces whose edge lengths $l$ are maintained within the range $[l_{min}, l_{max}]$ with a local mesh adaptation method. The minimum edge length $l_{min}$ is a model parameter, whereas $l_{max} = 3l_{min}$, a value that works well in most practical applications, is automatically set. When the length of an edge exceeds $l_{max}$, the local mesh adaptation method splits it in two (Supplementary Fig. 2a), adding one node and two faces to the mesh. The two nodes constituting the divided edge transfer a third of their momentum to the newly created middle node to ensure momentum continuity. When an edge shrinks to a length below $l_{min}$, it is collapsed into a node whose new momentum is the sum of the merged nodes (Supplementary Fig. 2b). This merging process eliminates one node and two faces from the cell mesh. To prevent triangles with vanishing area, this operation is allowed only when the two nodes to be merged share exactly two other nodes among those connected to them through edges.

Triangular faces with high isoperimetric ratios can be a source of numerical instability. An edge swap operation prevents their formation. First, the quality score $S_f = 36A_f/\sqrt{3}P_f^2$ of each face $f$ is computed, where $P_f$ is its perimeter and $A_f$ its area. Undesirable faces with high isoperimetric ratios have scores tending to zero, whereas $S_f = 1$ for equilateral triangles. Faces with $S_f < 0.3$ are eliminated by an edge swap operation (Supplementary Fig. 2c) that locally reconnects mesh nodes, but leaves them otherwise unaffected.

**Initial triangulation.** A flexible triangulation algorithm ensures that simulations are initialized with meshes that respect the edge length bounds (Supplementary Fig. 1a). The procedure takes an initial geometry of the tissue as input, with cell meshes that are not necessarily triangular yet, in the widely used VTK format[70]. The cell surfaces are then individually sampled with the Poisson disk sampling algorithm[71] (Supplementary Fig. 1b) with a minimal point separation of $l_{min}$. The ball pivoting algorithm[72,73] then separately re-triangulates the surface of each cell on the basis of its Poisson point cloud (Supplementary Fig. 1c). The resulting meshes have $l \geq l_{min}$, rarely exceeding $l_{max}$. Edges with $l > l_{max}$ are removed before the simulation starts with the edge division operation described above.

**Cell division.** Cells are divided on the basis of a volume threshold, that is, if $V > V_{max}$. They are bisected by a plane running through their centroid, whose orientation can depend on the cell type. The orientation

is either random or perpendicular to the cell's longest axis, as given by the eigenvector belonging to the smallest eigenvalue of its covariance matrix. During division, the cutting planes are re-triangulated in a manner respecting the edge length bounds. On the untriangulated region of the daughter cells, points are first sampled with the Poisson point cloud algorithm[71], and are then connected with the two-dimensional Delaunay triangulation algorithm. This method avoids a retriangulation of the parts of the cell surface inherited from the mother cell.

### Cell volume and area calculation

The cell volume is calculated with a three-dimensional variant of the shoelace formula:

$$V = \frac{1}{6}\left|\sum_{f\in\mathcal{M}} \det\begin{bmatrix} | & | & | \\ \mathbf{r}_i & \mathbf{r}_j & \mathbf{r}_k \\ | & | & | \end{bmatrix}\right|,$$

where $\mathbf{r}_i$, $\mathbf{r}_j$ and $\mathbf{r}_k$ are the nodal positions of face $f$ (Fig. 1c). The summation runs over all the triangular faces of the cell mesh $\mathcal{M}$. The cell surface area is obtained by summing the areas of its faces:

$$A = \sum_{f\in\mathcal{M}} A_f = \sum_{f\in\mathcal{M}} \frac{1}{2}\left\|\mathbf{n}_f\right\|,$$

where $\mathbf{n}_f = (\mathbf{r}_j - \mathbf{r}_i) \times (\mathbf{r}_k - \mathbf{r}_i)$ is the unnormalized outward normal of face $f$.

### Time integration

SimuCell3D offers two modes of time propagation, solving either the dynamic or overdamped equations of motion for the nodal positions $\mathbf{r}_i$,

$$m\ddot{\mathbf{r}}_i + \zeta\dot{\mathbf{r}}_i = \mathbf{f}_i.$$

The nodal mass $m$ is obtained from $V$ and mass density $\rho$ as $m = \rho V/N_n$, where $N_n$ is the total number of nodes in the cell mesh. $\mathbf{f}_i$ is the nodal force vector (specified below) and $\zeta$ the viscous damping coefficient. The first mode resolves elastic oscillations, making it suited for phenomena on short timescales. The nodal positions $\mathbf{r}_i$ and linear momenta $\mathbf{p}_i = m\mathbf{r}_i$ are integrated with the semi-implicit Euler scheme:

$$\mathbf{p}_i \leftarrow \mathbf{p}_i + \Delta t\left(\mathbf{f}_i - \zeta\mathbf{p}_i/m\right),$$
$$\mathbf{r}_i \leftarrow \mathbf{r}_i + \Delta t\, \mathbf{p}_i/m,$$

where $\Delta t$ is a fixed time increment. The second mode neglects inertial effects ($m\ddot{\mathbf{r}}_i = 0$) and is therefore suitable for systems dominated by viscous relaxation toward a quasistatic equilibrium. The overdamped equations of motion are solved with the forward Euler scheme:

$$\mathbf{r}_i \leftarrow \mathbf{r}_i + \Delta t\, \mathbf{f}_i/\zeta.$$

The simulations presented in Figs. 1, 3 and 4 were solved with the dynamic model. Simulation snapshots are written at regular time intervals in VTK format for post-processing and visualization in ParaView (Kitware).

### Nodal forces

The total conservative nodal forces $\mathbf{f}_i$ derive from the cell potential energy (equation (1)) and the intercellular interaction model. They are given by the sum of the surface tension forces, $\mathbf{f}_{s,i}$, the membrane area elasticity forces, $\mathbf{f}_{m,i}$, pressure forces exerted by the cytoplasm, $\mathbf{f}_{p,i}$, the bending forces, $\mathbf{f}_{b,i}$, and contact forces due to adhesion and steric repulsion, $\mathbf{f}_{c,i}$:

$$\mathbf{f}_i = \mathbf{f}_{s,i} + \mathbf{f}_{m,i} + \mathbf{f}_{p,i} + \mathbf{f}_{b,i} + \mathbf{f}_{c,i}.$$

Each of these contributions is detailed in the following paragraphs.

**Surface tension.** The surface tension force is given by the negative gradient of the surface tension energy with respect to the nodal position. Since the position of node $i$ affects the areas of only the set of faces $\mathcal{F}_i$ sharing this node, it is given by

$$\mathbf{f}_{s,i} = -\sum_{f\in\mathcal{F}_i} \gamma_f \nabla_i A_f,$$

where $\gamma_f$ is the surface tension of face $f$. For triangles with nodes $i$, $j$, $k$ oriented clockwise (Fig. 1c), the area gradient reads

$$\nabla_i A_f = \frac{1}{2}\hat{\mathbf{n}}_f \times (\mathbf{r}_k - \mathbf{r}_j),$$

where $\hat{\mathbf{n}}_f = \mathbf{n}_f / \left\|\mathbf{n}_f\right\|$, is the normalized face normal vector.

**Membrane area elasticity.** Similarly, the membrane force is obtained by taking the gradient of the cell membrane area energy with respect to $\mathbf{r}_i$:

$$\mathbf{f}_{m,i} = -\frac{k_a}{A_0}\left(\frac{A}{A_0} - 1\right)\sum_{f\in\mathcal{F}_i} \nabla_i A_f.$$

$A_0$ is coupled to $V_0$ via $A_0 = \sqrt[3]{Q_0 V_0^2}$, where $Q_0$ is the target isoperimetric ratio of the cell, which can be set by the user. For a sphere, $Q_0 = 36\pi \approx 113$.

**Pressure.** The cell-internal net pressure generated by the cytoplasm reads

$$p = \frac{\mathrm{d}W}{\mathrm{d}V} = -K \ln\frac{V}{V_0},$$

where $W = -KV[\ln(V/V_0) - 1]$ is the work associated with a deviation of $V$ from its reference value $V_0$. To model cell growth, $V_0$ can evolve over time according to prescribed growth laws, such as the linear form $\mathrm{d}V_0/\mathrm{d}t = g$, where $g$ is a constant volumetric growth rate that can vary from cell to cell. If desired, the pressure difference between the cell cytoplasm and the external medium can be capped at a predefined threshold $p_{max}$, that is, $p \leftarrow \min\{p, p_{max}\}$. The pressure force exerted on a subset of the cell surface $\mathcal{S} \subset \partial\Omega$ (where $\Omega$ is the cell domain) is given by

$$\mathbf{f}_{p,\mathcal{S}} = p\int_{\mathcal{S}} \hat{\mathbf{n}}\,\mathrm{d}S.$$

If the subset $\mathcal{S}$ of the cell surface is planar, like the triangular faces $f$ used to discretize the cell geometry, this simplifies to

$$\mathbf{f}_{p,f} = pA_f\hat{\mathbf{n}}_f.$$

The pressure force applied on each node of the cell mesh therefore follows as

$$\mathbf{f}_{p,i} = \frac{1}{3}\sum_{f\in\mathcal{F}_i} \mathbf{f}_{p,f}.$$

**Membrane bending.** The contribution of bending to the cell potential energy can be approximated with the discrete bending energy[74]

$$U_b \approx \sum_{(i,j)} \bar{k}_b \frac{\left\|\mathbf{e}_{ij}\right\|^2}{A_{ij}}\left(2\cos\frac{\theta_{ij}}{2}\right)^2$$

in which the sum runs over all pairs of nodes $(i, j)$ of the surface mesh connected by an edge. Each edge connects two faces $a$, $b$ that form a diamond region composed of four nodes $i$, $j$, $k$, $l$ (Fig. 1c).

$\bar{k}_\mathrm{b} = (k_{\mathrm{b},a} + k_{\mathrm{b},b})/2$ is the average bending stiffness of the faces $a$ and $b$, $\mathbf{e}_{ij} = \mathbf{r}_j - \mathbf{r}_i$ the vector pointing from node $i$ to $j$, $A_{ij} = A_a + A_b$ the sum of the two face areas and $\theta_{ij}$ the dihedral angle between the two faces:

$$\theta_{ij} = -\mathrm{sgn}\,(\hat{\mathbf{n}}_a \cdot \mathbf{e}_{il})\arccos\,(-\hat{\mathbf{n}}_a \cdot \hat{\mathbf{n}}_b)\,.$$

The sign of the dot product between the normal of face $a$ ($\hat{\mathbf{n}}_a$) and the vector $\mathbf{e}_{il}$ is used to distinguish between concave and convex hinges. The bending forces resulting from this discrete bending energy can be calculated independently for each of the four nodes, $q \in \{i, j, k, l\}$, as

$$\mathbf{f}_{\mathrm{b},q} = 2k_\mathrm{b}\left[\frac{\left\|\mathbf{e}_{ij}\right\|^2}{A_{ij}}\sin\theta_{ij}\,\nabla_q\theta_{ij} - (1+\cos\theta_{ij})\nabla_q\left(\frac{\left\|\mathbf{e}_{ij}\right\|^2}{A_{ij}}\right)\right].$$

For the gradients $\nabla_q\theta_{ij}$ and $\nabla_q(\|\mathbf{e}_{ij}\|^2/A_{ij})$ we refer the interested reader to ref. 74. The total bending force at node $i$, $\mathbf{f}_{\mathrm{b},i}$, then follows as the sum of bending forces over all diamond regions involving that node.

**Intercellular contacts.** SimuCell3D offers two different contact models that vary in their methods of exchanging contact forces between adjacent cells, but in the current version it does not take friction into account. (For possible ways to include frictional effects, see for example refs. 43,45.) The first model connects adjacent pairs of faces $\{f_a, f_b\}$ with elastic springs, while the second tightly couples pairs of adjacent nodes $\{n_a, n_b\}$.

The spring-based model applies contact forces on pairs of adjacent faces with a magnitude based on the signed distance $d_{ab} = \mathrm{sgn}\,(\mathbf{z}\cdot\hat{\mathbf{n}}_a)\,\|\mathbf{z}\|$ between the two mesh elements, where $\hat{\mathbf{n}}_a$ is the unit normal of face $a$ and $\mathbf{z}$ is the shortest vector between the two mesh elements. A contact stress is then calculated with the piecewise expression

$$\sigma_{ab} = \begin{cases} \xi d_{ab} & \text{if } d_{ab} \in [-c, 0) \\ \omega d_{ab} & \text{if } d_{ab} \in [0, c/2) \\ \omega(c - d_{ab}) & \text{if } d_{ab} \in [c/2, c] \\ 0 & \text{otherwise} \end{cases}. \qquad (2)$$

When two neighboring cells interpenetrate, $d_{ab}$ is negative, and the contact stress is repulsive. On the other hand, when $d_{ab}$ is positive, the contact stress is adhesive. In this regime, the contact model follows a bilinear traction–separation law (Supplementary Fig. 1). The contact stress $\sigma_{ab}$ thus obtained is translated into a force by integrating the contact stress over the contact surface area $A_{ab}$:

$$\mathbf{f}_{ab} = \mathrm{sgn}\,(\mathbf{z}\cdot\hat{\mathbf{n}}_a)\,A_{ab}\sigma_{ab}\frac{\mathbf{z}}{\|\mathbf{z}\|}.$$

$A_{ab} = \min\{A_a, A_b\}$ if the contact forces are computed between pairs of faces $\{f_a, f_b\}$, whereas $A_{ab} = A_a$ if the contact forces are calculated between pairs of faces and vertices $\{f_a, v_b\}$. In the first case, the force is linearly distributed to the nodes of face $a$, $\{i, j, k\}$, and the nodes of face $b$, $\{l, m, n\}$:

$$\mathbf{f}_{\mathrm{c},i} = \alpha_a\mathbf{f}_{ab}, \mathbf{f}_{\mathrm{c},j} = \beta_a\mathbf{f}_{ab}, \mathbf{f}_{\mathrm{c},k} = \lambda_a\mathbf{f}_{ab},$$
$$\mathbf{f}_{\mathrm{c},l} = -\alpha_b\mathbf{f}_{ab}, \mathbf{f}_{\mathrm{c},m} = -\beta_b\mathbf{f}_{ab}, \mathbf{f}_{\mathrm{c},n} = -\lambda_b\mathbf{f}_{ab}.$$

$(\alpha_a, \beta_a, \lambda_a)$ and $(\alpha_b, \beta_b, \lambda_b)$ are the barycentric coordinates of the closest points of approach on faces $a$ and $b$, respectively.

The second contact model eliminates the need for a finite $\omega$ by establishing a tight coupling between node pairs $\{n_a, n_b\}$ whose distance is smaller than the contact cutoff $c$. The two nodes are relocated to their average location $(\mathbf{r}_a + \mathbf{r}_b)/2$, and the forces and momenta acting on each node are transmitted to its partner such that both nodes subsequently follow the same dynamics: $\mathbf{f}_i \leftarrow (\mathbf{f}_a + \mathbf{f}_b)/2$ and $\mathbf{p}_i \leftarrow (\mathbf{p}_a + \mathbf{p}_b)/2$, $i = a, b$.

To allow two adjacent cells to detach from each other, node pairs are coupled only if the local mean curvature of both cell surfaces lies below the threshold $H_{\max}$ (Table 2). Coupled node pairs are redetermined in each time step, and each node is allowed to be coupled to no more than one other node.

### Reporting summary

Further information on research design is available in the Nature Portfolio Reporting Summary linked to this article.

## Data availability

The raw data generated as part of this study are publicly available and can be downloaded at https://u.ethz.ch/7Taih (ref. 75). Source data are provided with this paper.

## Code availability

SimuCell3D is open source and freely available as a public git repository at https://git.bsse.ethz.ch/iber/Publications/2024_runser_simucell3d under the three-clause BSD license[76].

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

## Acknowledgements

We thank F. Lampart for providing the prostate organoid images presented in Fig. 2f–h. This work was partially funded by SNF Sinergia grant CRSII5_170930.

## Author contributions

Concept: R.V., D.I. Model and algorithm development: S.R., R.V. Implementation: S.R. Numerical simulations: S.R. Figures: S.R. Writing: S.R., R.V., D.I.

## Funding


## Competing interests

The authors declare no competing interests.

## Additional information


**Correspondence and requests for materials** should be addressed to Dagmar Iber.


nature portfolio



# Reporting Summary


## Statistics

For all statistical analyses, confirm that the following items are present in the figure legend, table legend, main text, or Methods section.

| n/a | Confirmed | |
|---|---|---|
| ☐ | ☒ | The exact sample size ($n$) for each experimental group/condition, given as a discrete number and unit of measurement |
| ☐ | ☒ | A statement on whether measurements were taken from distinct samples or whether the same sample was measured repeatedly |
| ☒ | ☐ | The statistical test(s) used AND whether they are one- or two-sided<br>*Only common tests should be described solely by name; describe more complex techniques in the Methods section.* |
| ☒ | ☐ | A description of all covariates tested |
| ☒ | ☐ | A description of any assumptions or corrections, such as tests of normality and adjustment for multiple comparisons |
| ☐ | ☒ | A full description of the statistical parameters including central tendency (e.g. means) or other basic estimates (e.g. regression coefficient) AND variation (e.g. standard deviation) or associated estimates of uncertainty (e.g. confidence intervals) |
| ☒ | ☐ | For null hypothesis testing, the test statistic (e.g. $F$, $t$, $r$) with confidence intervals, effect sizes, degrees of freedom and $P$ value noted<br>*Give P values as exact values whenever suitable.* |
| ☒ | ☐ | For Bayesian analysis, information on the choice of priors and Markov chain Monte Carlo settings |
| ☒ | ☐ | For hierarchical and complex designs, identification of the appropriate level for tests and full reporting of outcomes |
| ☒ | ☐ | Estimates of effect sizes (e.g. Cohen's $d$, Pearson's $r$), indicating how they were calculated |

*Our web collection on statistics for biologists contains articles on many of the points above.*

## Software and code

Policy information about availability of computer code

| | |
|---|---|
| Data collection | The code of our program is publicly available under the 3-clauses BSD license in the following Git repository: https://git.bsse.ethz.ch/iber/Publications/2024_runser_simucell3d. The python scripts used to collect the simulation results can be freely accessed in the following OpenBis repository: https://u.ethz.ch/7Taih |
| Data analysis | The python scripts used to analyze the simulation results can also be freely accessed in the following OpenBis repository: https://u.ethz.ch/7Taih |

For manuscripts utilizing custom algorithms or software that are central to the research but not yet described in published literature, software must be made available to editors and reviewers. We strongly encourage code deposition in a community repository (e.g. GitHub). See the Nature Portfolio guidelines for submitting code & software for further information.

## Data

Policy information about availability of data

All manuscripts must include a data availability statement. This statement should provide the following information, where applicable:

- Accession codes, unique identifiers, or web links for publicly available datasets
- A description of any restrictions on data availability
- For clinical datasets or third party data, please ensure that the statement adheres to our policy

The datasets generated in this study are publicly available in the following OpenBis repository: https://u.ethz.ch/7Taih

## Human research participants

Policy information about studies involving human research participants and Sex and Gender in Research.

| | |
|---|---|
| Reporting on sex and gender | N/A |
| Population characteristics | N/A |
| Recruitment | N/A |
| Ethics oversight | N/A |

Note that full information on the approval of the study protocol must also be provided in the manuscript.

# Field-specific reporting

Please select the one below that is the best fit for your research. If you are not sure, read the appropriate sections before making your selection.

☒ Life sciences ☐ Behavioural & social sciences ☐ Ecological, evolutionary & environmental sciences

For a reference copy of the document with all sections, see nature.com/documents/nr-reporting-summary-flat.pdf

# Life sciences study design

All studies must disclose on these points even when the disclosure is negative.

| | |
|---|---|
| Sample size | Sample sizes were not predetermined using any statistical method. The number of cells simulated in Figures 3 and 4 was deemed sufficient, as continuous morphological changes in their shapes, as well as in tissue architecture, were observable when varying the cellular parameters (Fig. 3e-f and Fig. 4d-g). |
| Data exclusions | No data was excluded from the analyses. |
| Replication | The simulations presented in this study are deterministic in nature. They are therefore fully reproducible and do not need to be replicated. |
| Randomization | Randomization is not relevant to this study since the simulations presented are deterministic in nature. |
| Blinding | Blinding is not relevant to this study since the simulations presented are deterministic in nature. |

# Reporting for specific materials, systems and methods

We require information from authors about some types of materials, experimental systems and methods used in many studies. Here, indicate whether each material, system or method listed is relevant to your study. If you are not sure if a list item applies to your research, read the appropriate section before selecting a response.

## Materials & experimental systems

| n/a | Involved in the study | |
|---|---|---|
| ☒ | ☐ | Antibodies |
| ☒ | ☐ | Eukaryotic cell lines |
| ☒ | ☐ | Palaeontology and archaeology |
| ☒ | ☐ | Animals and other organisms |
| ☒ | ☐ | Clinical data |
| ☒ | ☐ | Dual use research of concern |

## Methods

| n/a | Involved in the study | |
|---|---|---|
| ☒ | ☐ | ChIP-seq |
| ☒ | ☐ | Flow cytometry |
| ☒ | ☐ | MRI-based neuroimaging |