## [Peer Review File · Nature Computational Science]

Peer Review Information

Journal: Nature Computational Science

Manuscript Title: SimuCell3D: 3D Simulation of Tissue Mechanics with Cell Polarization

Corresponding author name(s): Professor Dagmar Iber

Editorial Notes:

Reviewer Comments & Decisions:

Decision Letter, initial version:

Date: 24th October 23 16:20:52

Last Sent: 24th October 23 16:20:52

Triggered By: Fernando Chirigati

From: fernando.chirigati@us.nature.com

To: dagmar.iber@bsse.ethz.ch

BCC: fernando.chirigati@us.nature.com

Subject: Decision on Nature Computational Science manuscript NATCOMPUTSCI-23-0381A-Z

Message: ** Please ensure you delete the link to your author homepage in this e-mail if you wish to forward it to your co-authors. **

Dear Professor Iber,

Your manuscript "3D Simulation of Tissue Mechanics with Cell Polarization" has now been seen by 4 referees, whose comments are appended below. Please note that Referees #3 and #4 jointly reviewed the paper.

You will see that while they find your work of interest, they have raised points that need to be addressed before we can make a decision on publication.

Naturally, we will need you to address **all** of the points raised. While we ask you to address all of the points raised, the following points need to be substantially worked on:

- The friction model does not take friction between the cells into account. Please discuss this issue clearly.

- Please provide more explanation on whether or not the contact stiffness between the cells or the parameters with respect to surface tension are realistic.
- Regarding the lumen-based system, please discuss whether or not any ECM on the apical side, or an apical actin ring, is taken into account.
- With regard to the adhesion model, please show whether or not it scales consistently with the resolution of the cell triangulation.
- Please indicate or demonstrate how the lack of stochastic forces in the simulations affects the results.
- The literature review section lacks enough discussion on previous cell-based methods for tissue simulation and should be elaborated.

In addition, please note that our editorial team was not able to access the code. Please make sure that the provided link works, and that all of the code and data necessary to reproduce the results of the paper is made available. If the paper is eventually accepted for publication, please note that you will be required to also deposit all of the code and data in a DOI-minting repository, in addition to making everything available on GitHub or GitLab.

Please use the following link to submit your revised manuscript and a point-by-point response to the referees' comments (which should be in a separate document to any cover letter):

[REDACTED]

** This url links to your confidential homepage and associated information about manuscripts you may have submitted or be reviewing for us. If you wish to forward this e-mail to co-authors, please delete this link to your homepage first. **

To aid in the review process, we would appreciate it if you could also provide a copy of your manuscript files that indicates your revisions by making use of Track Changes or similar mark-up tools. Please also ensure that all correspondence is marked with your Nature Computational Science reference number in the subject line.

In addition, please make sure to upload a Word Document or LaTeX version of your text, to assist us in the editorial stage.

To improve transparency in authorship, we request that all authors identified as 'corresponding author' on published papers create and link their Open Researcher and Contributor Identifier (ORCID) with their account on the Manuscript Tracking System (MTS), prior to acceptance. ORCID helps the scientific community achieve unambiguous attribution of all scholarly contributions. You can create and link your ORCID from the home page of the MTS by clicking on 'Modify my Springer Nature account'. For more information please visit please visit www.springernature.com/orcid.

We hope to receive your revised paper within three weeks. If you cannot send it within this time, please let us know.

Best,

Fernando

--

Fernando Chirigati, PhD
Chief Editor, Nature Computational Science
Nature Portfolio

Reviewers comments:

Reviewer #1 (Remarks to the Author):

This paper presents a tissue simulation software based on agent-based technology whereby the cells are represented as highly deformable objects ("Deformable Cell Model", DCM).

The DCM is not a new methodology, and many aspects, as presented in this paper, are state-of-the-art and already described in earlier works. Here, the authors present a software package that they claim can simulate several 1000 to 100,000 cells in a time frame of 1 day of simulation time. This result is impressive, as it may increase the overall interest in the method significantly, which is now somewhat plagued by the large computational times.

Overall, the paper is well written and I find the results interesting and partially convincing. In this regard, some more explanation on the results is necessary and the method descriptions lacks some detail on several occasions.

My most important comments are:

1. Regarding the computational time diagram: Is this result obtained for cells each containing a nucleus inside? Furthermore, despite the claim of being able to simulate growing systems up to 100,00 cells, there is no clear visual proof. Can the authors give more detail which system they actually simulate to achieve this?
2. In my opinion, the friction model is somewhat simplified because it does not take friction between the cells into account. As a result, pure shearing effects between cells are not captured in the model. This simplification also greatly simplifies the system to solve and probably explains why the authors are capable of simulating such high cell numbers. I think the authors should clearly discuss this issue.
3. The large timestep (1000 s in the overdamped case) also explains why the simulations can handle such large cell numbers. However the maximum timestep that can be used in a stable simulation is usually determined by several parameters, for example stiffness values. I am bit surprised that such a large timesteps can be used in this DCM. Can the authors give more explanation on whether the contact stiffnesses between the cells or the parameters with respect to surface tension are realistic? What are the limitations here? Are the results sufficiently independent of the used timestep?
4. Regarding the lumen-based system. Did the authors take any ECM on the apical side, or an apical actin ring into account? This is often observed in organoids. Furthermore I have some doubts about the results in Figure 3 in the sense that cells are not dividing. Cells dividing will create large mechanical perturbations and as such I am not sure whether the observed structures (or the phase diagram Fig 3e) would

be the same after several cell divisions.

5. I am puzzled with how the authors achieved to simulate a cell splitting up almost completely in two parts (Fig 4c). The two cell parts are apparently only separated with a cord. Where does this cord come from? Can the authors be more clear on whether this is an emerging effect of the remeshing? I am a bit doubtful whether this splitting effect will not introduce artifacts in the simulations. In principle, a more in-depth analysis of the remeshing scheme should be present, for example by showing that remeshing algorithm does not increase or decreasing the energy for one cell.

6. With regard to the adhesion model, I have my doubts whether it scales consistently with the resolution of the cell triangulation. In my opinion the authors should prove that experiments that measure contact angles between cells, or pull off forces between cells, can be reproduced with the model. In the work by Smeets et al. (e.g. <https://www.sciencedirect.com/science/article/pii/S0006349523002680#fig4>) this has been investigated thoroughly.

7. Stochastic forces may play an important role in cellular systems. Surprisingly there seems to be no stochastic components (due to thermal fluctuations) in the simulations. Is there any reason for that? Can the authors indicate how this affects the simulation results?

Reviewer #1 (Remarks on code availability):

I had no time to review the code. Note that code for such models can be very large, and it may take a long time to review and test them thoroughly.

Reviewer #2 (Remarks to the Author):

This manuscript represents promising work in modelling the mechanics of large assemblies of cells in 3D. As the authors note, it accounts for growth, proliferation, extracellular matrix, fluid cavities, nuclei, and non-uniform mechanical properties of polarised epithelia at subcellular resolution. It therefore is relevant to spheroids, vesicles, sheets, tubes, and other tissue structures whose geometries can be obtained from microscopy images and their mechanics can be modelled. The authors also have presented two simulations of 3D many-cell structures that are impressive: the formation and maintenance of layered epithelia, and cellular organization in a "pseudostratified" epithelium. The computational treatment is detailed and uses a range of techniques in triangulation, mesh refinement and rezoning. The mechanics is treated by nodal force balance over the quasi-static, second- and first-order (over-damped) dynamic regimes. The forces themselves are obtained as the gradients of an energy functional accounting for area, volume and curvature elasticities.

My opinion, however, is that to qualify for Nature Comp Sci, work of this type needs to be either (a) of much greater scope drawing from a range of multiscale of multiphysics methods (for example using molecular, particle scale and continuum models or coupling chemistry, transport and mechanics in cells), or (b) the culmination of a series of focused earlier works, each of them going into significant depth in different aspects (the foundational physics, numerical methods, a few case studies...). The current manuscript does fall into either of these classes (a) or (b). It does, however fulfill the role of a manuscript representing a first step along Path b above. For this reason, I suggest that it may be more suitable for a forum in

computational cell biology.

Reviewer #2 (Remarks on code availability):

No, but from the examples presented here, I expect it works as advertised.

Reviewer #3 (Remarks to the Author):

The paper "SimuCell3D: 3D Simulation of Tissue Mechanics with Cell Polarization" is a very well structured and written article.

Here, the authors present a novel simulation implementation for individual cell-based modelling, which can produce (computationally) efficiently large tissues in 3D, representative of the real biological processes. The software is open-source, and to be made available to the public upon publication. The authors detail how their software is capable of describing subcellular resolution, growth, proliferation, extracellular matrices, fluid cavities, cell nuclei and cell polarity. They then present how their implementation can easily model spheroids, vesicles, epithelial sheets and tubes, as well as monolayer, pseudostratified and stratified tissue structures. The basis for the model implementation is class of Deformable Cell Model, implemented in 3D.

I have a few minor comments. This article was a pleasure to read, and following these comments being addressed, I would recommend the article for publication.

Comment 1:

The authors may wish to include a reference to the 2023 Advanced Materials paper by Schamberger et. al. which will emphasize this paper's relevance and usefulness even more.

Comment 2:

The literature review section lacks enough discussion on previous cell-based methods for tissue simulation.

Comment 3:

The statement:

"The high level of spatio-temporal details of cell-based models, however, entails a substantial computational cost which forces them to a tradeoff between the number of cells they can simulate and the spatial resolution of their representation."

Is made without any discussion or reference – a relevant reference or explanation should assist this statement, as the authors seem to provide a counter example to this statement in this very paper.

Comment 4:

Similar justification is needed for the statement:

"vertex models are unsuitable for tissues that possess complex cell shapes or undergo phenomena such as cell extrusion or lumenogenesis."

Comment 5:

On Page 3, the statement is made:

“SimuCell3D overcomes the classical trade-off that has so far constrained cell-based models between their resolutions and the number of cells they can simulate.”

Without providing reasoning or justification.

Comment 6:

There is a word (one/us/...) missing in the sentence:

“In addition, our program natively allows ____ to represent intra...”

Comment 7:

The statement:

“which preserves numerical stability of the program even under large cell deformations. ”

Is made, but there is no discussion about numerical stability. Please expand upon this point. Is it sensitive to time step, and what type of stability is being describe?

Comment 8:

Following Equation 1, please state that the “W” in dW/dV is work, as this may not be immediately obvious (it’s clear in the methods but not here).

Comment 9:

Please expand upon how the time complexity for the presented approach is found to be order $N_c^{4/3}$, as I would have thought it scaled with the number of nodes, rather than cells.

Comment 10:

On page 5, 4 lines from the bottom, I believe the words “study case” may need to be flipped?

Comment 11:

On page 6, I believe a word (one/us/...) may be missing:

“SimuCell3D also allows ____ to directly modulate the shapes ...”

Comment 12:

In the Methods section, can you please provide explanation why $l_{\max} = 3 \times l_{\min}$ is sufficient?

Is this arbitrary, and how sensitive are the results to this?

Comment 13:

On page 7, the final paragraph again describes sources of numerical stability. Please provide some further explanation as to why this occurs and how it may be prevented.

Comment 14:

Throughout the article, the jet colour pallet is used, except for Figure 4f. This jet colour pallet makes it difficult to observe small changes in variation, particularly non-monotonic variation (e.g. Figure 3e). Is it please possible the colour pallet used in Figure 4f is used throughout?

Comment 15:

I would really enjoy to see some simulation output videos to accompany the Vesicle, spheroid, sheet, and tube in Figure 1.

Comment 16:

I am curious to know if the software has been optimised for parallel computation across all 16 threads? If so, in the comparison to other method of Table 1, are these other implementations similarly optimised/parallel? If these are single thread implementations, I am curious to see how SimuCell3D compares on a single thread also.

Comment 17:

All equations should be punctuated. "." if the end of a sentence as for first equation on page 3 and 9 and a "," if its in the middle of a sentence as on page 8 (and others).

Comment 18:

There is a reversed bracket in one of the limits in Equation (2).

Reviewer #4 (Remarks to the Author):

The paper "SimuCell3D: 3D Simulation of Tissue Mechanics with Cell Polarization" is a very well structured and written article.

Here, the authors present a novel simulation implementation for individual cell-based modelling, which can produce (computationally) efficiently large tissues in 3D, representative of the real biological processes. The software is open-source, and to be made available to the public upon publication. The authors detail how their software is capable of describing subcellular resolution, growth, proliferation, extracellular matrices, fluid cavities, cell nuclei and cell polarity. The then present how their implementation can easily model spheroids, vesicles, epithelial sheets and tubes, as well as monolayer, pseudostratified and stratified tissue structures. The basis for the model implementation is class of Deformable Cell Model, implemented in 3D.

I have a some minor comments. This article was a pleasure to read, and following these comments being address, I would recommend the article for publication.

Comment 1:

The authors may wish to include a reference to the 2023 Advanced Materials paper by Schamberger et. al. which will emphasis this papers relevance and usefulness even more.

Comment 2:

The literature review section lacks enough discussion on previous cell-based methods for tissue simulation.

Comment 3:

The statement:

"The high level of spatio-temporal details of cell-based models, however, entails a substantial computational cost which forces them to a tradeoff between the number of cells they can simulate and the spatial resolution of their representation."

Is made without any discussion or reference – a relevant reference or explanation should assist this statement, as the authors seem to provide a counter example to this statement in this very paper.

Comment 4:

Similar justification is needed for the statement:

"vertex models are unsuitable for tissues that possess complex cell shapes or undergo phenomena such as cell extrusion or lumenogenesis."

Comment 5:

On Page 3, the statement is made:

"SimuCell3D overcomes the classical trade-off that has so far constrained cell-based models between their resolutions and the number of cells they can simulate."

Without providing reasoning or justification.

Comment 6:

There is a word (one/us/...) missing in the sentence:

"In addition, our program natively allows ____ to represent intra..."

Comment 7:

The statement:

"which preserves numerical stability of the program even under large cell deformations. "

Is made, but there is no discussion about numerical stability. Please expand upon this point. Is it sensitive to time step, and what type of stability is being describe?

Comment 8:

Following Equation 2, please state that the "W" in dW/dV is work, as this may not be immediately obvious.

Comment 9:

Please expand upon how the time complexity for the presented approach is found to be order $N_c^{4/3}$, as I would have thought it scaled with the number of nodes, rather than cells.

Comment 10:

On page 5, 4 lines from the bottom, I believe the words "study case" may need to be flipped?

Comment 11:

On page 6, I believe a word (one/us/...) may be missing:

"SimuCell3D also allows ____ to directly modulate the shapes ..."

Comment 12:

In the Methods section, can you please provide explanation why $l_{\max} = 3 \times l_{\min}$ is sufficient?

Is this arbitrary, and how sensitive are the results to this?

Comment 13:

On page 7, the final paragraph again describes sources of numerical stability. Please provide some further explanation as to why this occurs and how it may be prevented.

Comment 14:

Throughout the article, the jet colour pallet is used, except for Figure 4f. This jet colour pallet makes it difficult to observe small changes in variation, particularly non-monotonic variation (e.g. Figure 3e). Is it please possible the colour pallet used in Figure 4f is used throughout?

Comment 15:

I would really enjoy to see some simulation output videos to accompany the Vesicle, spheroid, sheet, and tube in Figure 1.

Comment 16:

I am curious to know if the software has been optimised for parallel computation across all 16 threads? If so, in the comparison to other method of Table 1, are these other implementations similarly optimised/parallel? If these are single thread implementations, I am curious to see how SimuCell3D compares on a single thread also.

Comment 17:

All equations should be punctuated. "." if the end of a sentence as for first equation on page 3 and 9 and a "," if its in the middle of a sentence as on page 8 (and others).

Comment 18:

There is a reversed bracket in one of the limits in Equation (2).

Author Rebuttal to Initial comments

Reviewers comments:

Reviewer #1 (Remarks to the Author):

This paper presents a tissue simulation software based on agent-based technology whereby the cells are represented as highly deformable objects (“Deformable Cell Model”, DCM).

The DCM is not a new methodology, and many aspects, as presented in this paper, are state-of-the-art and already described in earlier works. Here, the authors present a software package that they claim can simulate several 1000 to 100,000 cells in a time frame of 1 day of simulation time. This result is impressive, as it may increase the overall interest in the method significantly, which is now somewhat plagued by the large computational times.

Overall, the paper is well written and I find the results interesting and partially convincing. In this regard, some more explanation on the results is necessary and the method descriptions lacks some detail on several occasions.

Thank you.

My most important comments are:

1. Regarding the computational time diagram: Is this result obtained for cells each containing a nucleus inside? Furthermore, despite the claim of being able to simulate growing systems up to 100,000 cells, there is no clear visual proof. Can the authors give more detail which system they actually simulate to achieve this?

The benchmark simulation was started from a single spherical cell without a nucleus, which we now mention in the manuscript. This cell grew and divided upon reaching a predefined division volume. Its progeny followed the same pattern of growth and division. The whole patch of cells spontaneously adopted a spherical shape as it grew and proliferated. We have included a new video (Movie S1) to facilitate the understanding of the simulated process.

2. In my opinion, the friction model is somewhat simplified because it does not take friction between the cells into account. As a result, pure shearing effects between cells are not captured in the model. This simplification also greatly simplifies the system to solve and probably explains why the authors are capable of simulating such high cell numbers. I think the authors should clearly discuss this issue.

Indeed, in the current version, our model does not take frictional forces between cells into account. These forces could be included by following an implementation such as the one described in Van Liedekerke et al, 2019 [R1]. However, if the conservative and dissipative forces are not balanced, there is no need to solve a system of equations at each iteration. We would therefore not expect the performance of our program to be significantly affected by the calculation of these frictional forces. In a similar 2D model we are currently developing [R2], a simple friction model is included and has negligible impact on the

computational cost. We now explicitly state in the model description that friction forces between cells are not taken into account, referring to literature that demonstrates how this could be added in future versions.

[R1] Van Liedekerke P, et al, A quantitative high-resolution computational mechanics cell model for growing and regenerating tissues. *Biomechanics and Modeling in Mechanobiology* 19 (2020). doi: 10.1007/s10237-019-01204-7

[R2] Vetter R, et al, PolyHoop: Soft particle and tissue dynamics with topological transitions. arXiv:2307.15006. doi: 10.48550/arXiv.2307.15006

3. The large timestep (1000 s in the overdamped case) also explains why the simulations can handle such large cell numbers. However, the maximum timestep that can be used in a stable simulation is usually determined by several parameters, for example, stiffness values. I am bit surprised that such a large timesteps can be used in this DCM. Can the authors give more explanation on whether the contact stiffnesses between the cells or the parameters with respect to surface tension are realistic? What are the limitations here? Are the results sufficiently independent of the used timestep?

Most of the mechanical cell parameters were obtained from published experimental measurements (Table 2). The parameters relative to the contacts had to be obtained through calibration rounds, such that the cells adhere together but do not interpenetrate. To the best of our knowledge, there exists no experimental measurement of the adhesion energy between cells [R3] that we could compare with the adhesion strength values used in the simulations.

As stated in the manuscript, to obtain such a large number of cells we used a large growth rate and thus simulate the development of the tissue in an out-of-equilibrium regime. Note that in the overdamped case (pseudo-timestepping), the timestep Δt and the damping coefficient ζ appear only together as a single parameter, the ratio $\Delta t / \zeta$ (see section "Time integration" in the Methods), such that the absolute value of Δt bears no meaning. To avoid misunderstandings with the readership of our paper, we have changed the default parameters such that the timestep is the same for both time propagation methods (Table 2).

We tested the impact of the time step on the results presented in Figure 4C. After selecting a combination of surface tension and adherence strength values ($\tilde{\gamma} = 0.003, \tilde{\omega} = 2.75$) that yields physiological cell shapes, we ran simulations with time steps ranging from 4×10^{-9} to 1.4×10^{-7} . The results, summarized in the following figure, reveal no observable influence of the time step on the cell shapes.

[R3] Maître JL, et al, The role of adhesion energy in controlling cell–cell contacts. *Current Opinion in Cell Biology* 23 (2011). doi: 10.1016/j.ceb.2011.07.004

4. Regarding the lumen-based system. Did the authors take any ECM on the apical side, or an apical actin ring into account? This is often observed in organoids. Furthermore I have some doubts about the results in Figure 3 in the sense that cells are not dividing. Cells dividing will create large mechanical perturbations and as such I am not sure whether the observed structures (or the phase diagram Fig 3e) would be the same after several cell divisions.

We did not take the presence of an extracellular matrix (ECM) or adhesion belts on the apical side of the cells into account. As highlighted by the reviewer, these structures are ubiquitous in epithelial tissues and are recognized to significantly contribute to their structural integrity. We plan to incorporate these structures in a future version of our program and study how they impact the cohesion and stability of epithelial layers.

Regarding cell division, we share the viewpoint expressed by the reviewer. The occurrence of cell divisions or variability in individual cell growth rates could introduce substantial mechanical perturbations, potentially destabilizing the monolayer structure. However, given that the primary focus of Figure 3 is solely to illustrate how our model can be applied to investigate the influence of cell mechanical parameters on tissue configurations, we refrained from incorporating these additional factors. Instead, we opted for a simplified analysis to maintain clarity. Since our program is open-source, interested researchers can study the mentioned effects with it once it is published.

5. I am puzzled with how the authors achieved to simulate a cell splitting up almost completely in two parts (Fig 4c). The two cell parts are apparently only separated with a cord. Where does this cord come from? Can the authors be more clear on whether this is an emerging effect of the remeshing? I am a bit doubtful whether this splitting effect will not introduce artifacts in the simulations. In principle, a more in-dept analysis of the remeshing scheme should be present, for example by showing that remeshing algorithm does not increase or decreasing the energy for one cell.

This cord is a string of triangles with high aspect ratios which naturally appears during the simulation as the cell shrinks along its radial direction. Such shrinkage in the radial direction is independent of the local remeshing scheme and rather represents a physical effect of fluidic closed surfaces driven by surface tension, as simulated here. Like soap bubbles, they can separate in principle. But since our meshes stay homeomorphic to spheres, the separated parts remain connected by a cord with vanishing surface area. In a biological setting where the meshes represent cell membranes, for example, this may be unphysiological, which is why we indeed term this regime unphysiological in the manuscript.

As detailed in the "Local mesh adaptation" subsection of the method section, the kinetic energy is intrinsically conserved during local remeshing. Only intermittently, the remeshing operations slightly alter the cell volumes and surface areas in curved surface regions. However, the target volumes and areas are unaffected, such that the previous volumes and areas are elastically restored in subsequent timesteps, and no cytoplasm or membrane material is lost numerically. Spurious potential energy introduced by remeshing is of the order of the surface discretization and is dissipated away with damping. To the best of our knowledge, we are not aware of any simple local remeshing scheme that is exempt from similar behavior.

6. With regard to the adhesion model, I have my doubts whether it scales consistently with the resolution of the cell triangulation. In my opinion the authors should prove that experiments that measure contact angles between cells, or pull off forces between cells, can be reproduced with the model. In the work by Smeets et al. (e.g. <https://www.sciencedirect.com/science/article/pii/S0006349523002680#fig4>) this has been investigated thoroughly.

As requested, we have tested the impact of mesh resolution on our contact model based on elastic springs. To do so, we have simulated the equilibrium shape of a cell doublet in which the two cells have the same apical and lateral surface tensions in all simulations. The adhesion strength between the two cells was also kept the same in all simulations. We varied the resolution of the mesh from high ($l_{\min} = 6 \times 10^{-7}$ m, \cong 3,500 nodes per cell) to low ($l_{\min} = 1.8 \times 10^{-6}$ m, \cong 300 nodes per cell). The results of this test are presented in the new Supplementary Figure S4. We observe a variation of 10% in the contact angle after reducing the number of nodes per cell by a factor 3. Given the large reduction of nodes needed to observe such variation, we conclude that the results obtained with the contact model based on springs only weakly depend on the mesh resolution.

We thank the reviewer for bringing the work of Smeets et al. to our attention, which had escaped our literature review. We have now incorporated their findings into our manuscript. Smeets et al. developed a model where the contact angles between cells follow the Young–Dupré equation. While the Young–Dupré law provides an intuitive framework for modeling cell contacts, to the best of our knowledge, there exists no published experiment proving the validity of this law in biological tissues. On the contrary, recent work by Slováková et al. [R4] tends to demonstrate the opposite with zebrafish embryonic cells. Therefore, we believe that designing a contact model that follows the Young–Dupré equation is more a modeling choice than a physical necessity.

Despite this, we acknowledge the widespread use of the Young–Dupré law in modeling cell contacts [R5,R6], rendering it potentially valuable to future users of our software. Consequently, we developed a second adhesion model that conforms to this law (refer to subsection "Intercellular contacts" in the methods). We tested that this additional contact model follows the Young–Dupré law at different mesh resolutions with a cell doublet geometry (Fig S4a,b). We also verified that it reproduces the angles at tricellular junctions as predicted by the Young–Dupré equation (Fig S4c). Finally, we reproduced the internalization of cells as predicted by the theory developed in [R7].

[R4] Slováková J, et al. Tension-dependent stabilization of E-cadherin limits cell–cell contact expansion in zebrafish germ-layer progenitor cells. *Proceedings of the National Academy of Sciences* 119 (2022). doi: 10.1073/pnas.2122030119

[R5] Maître JL, et al. Adhesion Functions in Cell Sorting by Mechanically Coupling the Cortices of Adhering Cells. *Science* 338 (2012). doi: 10.1126/science.1225399

[R6] Maître JL, et al. Pulsatile cell-autonomous contractility drives compaction in the mouse embryo. *Nature Cell Biology* 17 (2015). doi: 10.1038/ncb3185

[R7] Maître JL, et al. Asymmetric division of contractile domains couples cell positioning and fate specification. *Nature* 536 (2016). doi: 10.1038/nature18958

7. Stochastic forces may play an important role in cellular systems. Surprisingly there seems to be no stochastic components (due to thermal fluctuations) in the simulations. Is there any reason for that ? Can the authors indicate how this affects the simulation results?

Indeed, the effect of surface tension fluctuation at the cell surfaces has recently been demonstrated to impact the mechanical properties of some tissues [R8]. The effect of these stochastic forces remains an active field of research that is beyond the scope of this paper. It is however worth noting that our program, with its new adhesion model based on node coupling, represents the cell geometries in 3D like the model introduced by Kim et al does in 2D. Readers interested in exploring the impact of stochastic tension fluctuations on the three-dimensional organization of tissues could therefore integrate this functionality into SimuCell3D by following the implementation presented by Kim et al.

[R8] Kim S, et al. Embryonic tissues as active foams. Nature Physics 17 (2021). doi: 10.1038/s41567-021-01215-1

Reviewer #1 (Remarks on code availability):

I had no time to review the code. Note that code for such models can be very large, and it may take a long time to review and test them thoroughly.

Reviewer #2 (Remarks to the Author):

This manuscript represents promising work in modelling the mechanics of large assemblies of cells in 3D. As the authors note, it accounts for growth, proliferation, extracellular matrix, fluid cavities, nuclei, and non-uniform mechanical properties of polarised epithelia at subcellular resolution. It therefore is relevant to spheroids, vesicles, sheets, tubes, and other tissue structures whose geometries can be obtained from microscopy images and their mechanics can be modelled. The authors also have presented two simulations of 3D many-cell structures that are impressive: the formation and maintenance of layered epithelia, and cellular organization in a "pseudostratified" epithelium. The computational treatment is detailed and uses a range of techniques in triangulation, mesh refinement and rezoning. The mechanics is treated by nodal force balance over the quasi-static, second- and first-order (over-damped) dynamic regimes. The forces themselves are obtained as the gradients of an energy functional accounting for area, volume and curvature elasticities.

My opinion, however, is that to qualify for Nature Comp Sci, work of this type needs to be either (a) of much greater scope drawing from a range of multiscale of multiphysics methods (for example using molecular, particle scale and continuum models or coupling chemistry, transport and mechanics in cells), or (b) the culmination of a series of focused earlier works, each of them going into significant depth in different aspects (the foundational physics, numerical methods, a few case studies...). The current manuscript does fall into either of these classes (a) or (b). It does, however fulfill the role of a manuscript representing a first step along Path b above. For this reason, I suggest that it may be more suitable for a forum in computational cell biology.

Reviewer #2 (Remarks on code availability):

No, but from the examples presented here, I expect it works as advertised.

The work outlined in this paper enhances the feature set, and also the performance of high-resolution cell-based models by several orders of magnitude. This makes it possible to simulate processes that were so far not amenable to cell-based modeling. We hope that the various upcoming applications of this model, which our group is currently pursuing, will convince Reviewer 2 of the broad potential of our program. For the present article format, we unfortunately have only limited room for biological applications.

Reviewer #3 (Remarks to the Author):

The paper “SimuCell3D: 3D Simulation of Tissue Mechanics with Cell Polarization” is a very well structured and written article.

Here, the authors present a novel simulation implementation for individual cell-based modelling, which can produce (computationally) efficiently large tissues in 3D, representative of the real biological processes. The software is open-source, and to be made available to the public upon publication. The authors detail how their software is capable of describing subcellular resolution, growth, proliferation, extracellular matrices, fluid cavities, cell nuclei and cell polarity. The then present how their implementation can easily model spheroids, vesicles, epithelial sheets and tubes, as well as monolayer, pseudostratified and stratified tissue structures. The basis for the model implementation is class of Deformable Cell Model, implemented in 3D.

I have a some minor comments. This article was a pleasure to read, and following these comments being address, I would recommend the article for publication.

Thank you.

Comment 1:

The authors may wish to include a reference to the 2023 Advanced Materials paper by Schamberger et. al. which will emphasis this papers relevance and usefulness even more.

We thank the reviewer for bringing this review to our attention, which highlights the requirements of a numerical approach such as the one we present in this paper. We have now incorporated a reference to it in the introduction.

Comment 2:

The literature review section lacks enough discussion on previous cell-based methods for tissue simulation.

We have extended the discussion of existing cell-based models by offering additional examples of their application. Regrettably, due to the limited word count allowed for this manuscript, we can only present the main types of cell-based models and provide a few application examples.

Comment 3:

The statement:

“The high level of spatio-temporal details of cell-based models, however, entails a substantial computational cost which forces them to a tradeoff between the number of cells they can simulate and the spatial resolution of their representation.”

Is made without any discussion or reference – a relevant reference or explanation should assist this statement, as the authors seem to provide a counter example to this statement in this very paper.

Osborne et al. [R9] have compared the computational performance of various types of cell-based models including center-based and vertex models. Their findings support the claim that the computational cost of these models positively correlates with their spatial resolution. We now cite their work in the manuscript.

[R9] Osborne J. M. , et al. Comparing individual-based approaches to modeling the self-organization of multicellular tissues. PLOS Computational Biology 13 (2017). doi: 10.1371/journal.pcbi.1005387

Comment 4:

Similar justification is needed for the statement:

“vertex models are unsuitable for tissues that possess complex cell shapes or undergo phenomena such as cell extrusion or lumenogenesis.”

We have rephrased this statement to make it hopefully less contentious. Vertex models cannot simulate complex cell shapes because they can only represent flat interfaces between adjacent cells. Real cells, on the other hand, can have strongly curved membranes, the representation of which can be important [R10]. Simulating phenomena such as tissue fusion or cell extrusions with vertex models is tedious because they require the implementation of mesh rearrangement operations named T2 and T3 transitions [R11]. To the best of our knowledge, there exists no 3D vertex model implementation that can perform these topological operations.

[R10] Perrone, M. C., et al. Non-straight cell edges are important to invasion and engulfment as demonstrated by cell mechanics model. Biomech. Model. Mechanobiol. 15 (2016). doi: 10.1007/s10237-015-0697-6.

[R11] Fletcher G. A. , et al. Vertex Models of Epithelial Morphogenesis. Biophysical Journal 106 (2014). doi: 10.1016/j.bpj.2013.11.4498

Comment 5:

On Page 3, the statement is made:

“SimuCell3D overcomes the classical trade-off that has so far constrained cell-based models between their resolutions and the number of cells they can simulate.”

Without providing reasoning or justification.

SimuCell3D is the first high-resolution cell-based model that has been proven capable of simulating thousands of cells in a short amount of time (Fig. 1E). We could only test two other deformable cell-based models (DCM) in similar conditions, namely CellSim3D and IAS. CellSim3D is a low-resolution DCM whereas IAS, like SimuCell3D, represents cells with a high geometrical resolution. Our program is as efficient as CellSim3D, despite its more complex geometric representation of cells, and is several orders of magnitude faster than IAS (Fig. 1E).

Comment 6:

There is a word (one/us/...) missing in the sentence:

“In addition, our program natively allows ____ to represent intra...”

We have rectified this error. Thank you for bringing it to our attention.

Comment 7:

The statement:

“which preserves numerical stability of the program even under large cell deformations. “

Is made, but there is no discussion about numerical stability. Please expand upon this point. Is it sensitive to time step, and what type of stability is being describe?

This statement primarily aimed to convey that the local remeshing scheme is designed to preserve the quality of triangular meshes. We have now revised it accordingly in the manuscript. The application of this scheme ensures that the meshes remain isotropic, meaning that they are made of triangles with similar edge lengths and areas. This isotropy enhances the numerical stability of mesh-based mechanical simulation programs, such as the one presented in this paper, by preventing the formation of triangles with areas close to zero. Such triangles would cause excessively large nodal forces that would require prohibitively small time steps to be integrated.

Comment 8:

Following Equation 1, please state that the “W” in dW/dV is work, as this may not be immediately obvious (it’s clear in the methods but not here).

We now explicitly state it in the manuscript.

Comment 9:

Please expand upon how the time complexity for the presented approach is found to be order $N_c^{4/3}$, as I would have thought it scaled with the number of nodes, rather than cells.

The average number of nodes per cell remains very stable throughout the simulations. Consequently, the number of nodes is directly proportional to the number of cells. The time complexity of the program therefore scales analogously with the number of nodes.

Comment 10:

On page 5, 4 lines from the bottom, I believe the words “study case” may need to be flipped?

Thank you very much for pointing out this error, we have rectified it.

Comment 11:

On page 6, I believe a word (one/us/...) may be missing:

“SimuCell3D also allows ___ to directly modulate the shapes ...”

Thank you again, we have corrected this.

Comment 12:

In the Methods section, can you please provide explanation why $l_{\max} = 3 \times l_{\min}$ is sufficient? Is this arbitrary, and how sensitive are the results to this?

The ratio between the minimum and maximum edge length defines the degree of isotropy of the mesh. As highlighted in response to comment 7, meshes with high isotropy contribute to improving the numerical stability of mechanical simulations. Consequently, this ratio should be set to a small value. However, it cannot be smaller than two, as edges generated during splitting operations would then immediately be merged since their lengths would be below the minimum threshold. This kind of behavior can propagate to the surrounding edges and lead to infinite loops of remeshing. We therefore arbitrarily set this ratio to 3, as it yields a good mesh isotropy without requiring too frequent mesh refinement operations. We have not observed any appreciable sensitivity of the results to this numerical ratio within reasonable ranges.

Comment 13:

On page 7, the final paragraph again describes sources of numerical stability. Please provide some further explanation as to why this occurs and how it may be prevented.

In the local mesh adaptation subsection of the methods, we indeed mention that triangles with small areas can be a source of numerical instabilities. As previously addressed in response to comment 7, this challenge is a common issue in mesh-based mechanical simulations, and it can be addressed by either reducing the time step or preventing the formation of such triangles through remeshing. In our specific case, triangles whose area approaches zero can pose a problem because the gradient of their areas with respect to their nodal degrees of freedom can take large values. Consequently, the surface tension and membrane elasticity forces can get large at the nodes of these triangles compared to the rest of the mesh. While this is of course physically correct, it is numerically undesirable, because it corresponds to a large

condition number, which requires unnecessarily small timesteps. Our program automatically avoids this numerical problem by keeping the mesh resolution relatively uniform.

Comment 14:

Throughout the article, the jet colour pallet is used, except for Figure 4f. This jet colour pallet makes it difficult to observe small changes in variation, particularly non-monotonic variation (e.g. Figure 3e). Is it please possible the colour pallet used in Figure 4f is used throughout?

As requested, we have modified the color map used in the different plots presented in this paper.

Comment 15:

I would really enjoy to see some simulation output videos to accompany the Vesicle, spheroid, sheet, and tube in Figure 1.

We have added a video to the manuscript, displaying the development of a spherical vesicle (Movie S2). For the other cases, we do not have such output videos, as the tissues were not simulated to emerge from simpler shapes, but were initialized already with the desired topology. We hope to study the emergence and dynamic transition between different tissue topologies in future work with SimuCell3D.

Comment 16:

I am curious to know if the software has been optimised for parallel computation across all 16 threads? If so, in the comparison to other method of Table 1, are these other implementations similarly optimised/parallel? If these are single thread implementations, I am curious to see how SimuCell3D compares on a single thread also.

SimuCell3D has been parallelized with OpenMP, which we now mention in the paper. We have analyzed the parallelization efficiency of our program in the case of an exponential growth simulation. The results are summarized in Figure S6 and indicate that our framework follows Amdahl's law with a parallel fraction of $P=55\%$ when it uses the spring-based contact model, and $P=66\%$ when it uses the new coupling contact model. With 16 threads, the contact model based on springs was running at 93% of its maximum theoretical speedup ($S_{max} = 1/(1 - P)$) in the benchmark of Fig 1E. The other implementations tested in this paper are also parallelized. CellSim3D has been designed to run in parallel on the GPU via CUDA, whereas IAS runs in parallel with OpenMPI on the CPU. The benchmark was performed with the same number of CPU threads (16) for SimuCell3D and IAS.

Comment 17:

All equations should be punctuated. "." if the end of a sentence as for first equation on page 3 and 9 and a "," if its in the middle of a sentence as on page 8 (and others).

As requested, we have punctuated all the equations.

Comment 18:

There is a reversed bracket in one of the limits in Equation (2).

These reversed brackets were meant to indicate that the upper limit of the range is not part of the interval. We have changed them for round parenthesis, which is another conventional notation for the same thing, to make it hopefully clearer.

Reviewer #4 (Remarks to the Author):

The paper “SimuCell3D: 3D Simulation of Tissue Mechanics with Cell Polarization” is a very well structured and written article.

Here, the authors present a novel simulation implementation for individual cell-based modelling, which can produce (computationally) efficiently large tissues in 3D, representative of the real biological processes. The software is open-source, and to be made available to the public upon publication. The authors detail how their software is capable of describing subcellular resolution, growth, proliferation, extracellular matrices, fluid cavities, cell nuclei and cell polarity. They then present how their implementation can easily model spheroids, vesicles, epithelial sheets and tubes, as well as monolayer, pseudostratified and stratified tissue structures. The basis for the model implementation is class of Deformable Cell Model, implemented in 3D.

I have a some minor comments. This article was a pleasure to read, and following these comments being address, I would recommend the article for publication.

Comment 1:

The authors may wish to include a reference to the 2023 Advanced Materials paper by Schamberger et. al. which will emphasis this papers relevance and usefulness even more.

Comment 2:

The literature review section lacks enough discussion on previous cell-based methods for tissue simulation.

Comment 3:

The statement:

“The high level of spatio-temporal details of cell-based models, however, entails a substantial computational cost which forces them to a tradeoff between the number of cells they can simulate and the spatial resolution of their representation.”

Is made without any discussion or reference – a relevant reference or explanation should assist this statement, as the authors seem to provide a counter example to this statement in this very paper.

Comment 4:

Similar justification is needed for the statement:

“vertex models are unsuitable for tissues that possess complex cell shapes or undergo phenomena such as cell extrusion or lumenogenesis.”

Comment 5:

On Page 3, the statement is made:

“SimuCell3D overcomes the classical trade-off that has so far constrained cell-based models between their resolutions and the number of cells they can simulate.”

Without providing reasoning or justification.

Comment 6:

There is a word (one/us/...) missing in the sentence:

“In addition, our program natively allows ___ to represent intra...”

Comment 7:

The statement:

“which preserves numerical stability of the program even under large cell deformations. “

Is made, but there is no discussion about numerical stability. Please expand upon this point. Is it sensitive to time step, and what type of stability is being describe?

Comment 8:

Following Equation 2, please state that the “W” in dW/dV is work, as this may not be immediately obvious.

Comment 9:

Please expand upon how the time complexity for the presented approach is found to be order $N_c^{4/3}$, as I would have thought it scaled with the number of nodes, rather than cells.

Comment 10:

On page 5, 4 lines from the bottom, I believe the words “study case” may need to be flipped?

Comment 11:

On page 6, I believe a word (one/us/...) may be missing:

“SimuCell3D also allows ___ to directly modulate the shapes ...”

Comment 12:

In the Methods section, can you please provide explanation why $l_{\max} = 3 \times l_{\min}$ is sufficient? Is this arbitrary, and how sensitive are the results to this?

Comment 13:

On page 7, the final paragraph again describes sources of numerical stability. Please provide some further explanation as to why this occurs and how it may be prevented.

Comment 14:

Throughout the article, the jet colour pallet is used, except for Figure 4f. This jet colour pallet makes it difficult to observe small changes in variation, particularly non-monotonic variation (e.g. Figure 3e). Is it please possible the colour pallet used in Figure 4f is used throughout?

Comment 15:

I would really enjoy to see some simulation output videos to accompany the Vesicle, spheroid, sheet, and tube in Figure 1.

Comment 16:

I am curious to know if the software has been optimised for parallel computation across all 16 threads? If so, in the comparison to other method of Table 1, are these other implementations similarly optimised/parallel? If these are single thread implementations, I am curious to see how SimuCell3D compares on a single thread also.

Comment 17:

All equations should be punctuated. "." if the end of a sentence as for first equation on page 3 and 9 and a "," if its in the middle of a sentence as on page 8 (and others).

Comment 18:

There is a reversed bracket in one of the limits in Equation (2).

Decision Letter, first revision:

Date: 5th February 24 15:15:42
Last Sent: 5th February 24 15:15:42
Triggered By: Fernando Chirigati
From: fernando.chirigati@us.nature.com
To: dagmar.iber@bsse.ethz.ch
CC: computacionalscience@nature.com
BCC: fernando.chirigati@us.nature.com
Subject: AIP Decision on Manuscript NATCOMPUTSCI-23-0381B
Message: Our ref: NATCOMPUTSCI-23-0381B

5th February 2024

Dear Dr. Iber,

Thank you for submitting your revised manuscript "SimuCell3D: 3D Simulation of Tissue Mechanics with Cell Polarization" (NATCOMPUTSCI-23-0381B). It has now been seen by the original referees and their comments are below. The reviewers find that the paper has improved in revision, and therefore we'll be happy in principle to publish it in Nature Computational Science, pending minor revisions to satisfy the referees' final requests and to comply with our editorial and formatting guidelines.

We are now performing detailed checks on your paper and will send you a checklist detailing our editorial and formatting requirements in about a week. Please **do not upload** the final materials and make any revisions until you receive this additional information from us. You are encouraged to start working on the final comments made by the referees, in particular related to the code (improved README, and a docker container to allow others with non-Linux machine to run the proposed resource and to improve usability) and presentation of results (better comparison with respect to computational efficiency).

TRANSPARENT PEER REVIEW

Nature Computational Science offers a transparent peer review option for original research manuscripts. We encourage increased transparency in peer review by publishing the reviewer comments, author rebuttal letters and editorial decision letters if the authors agree. Such peer review material is made available as a supplementary peer review file. **Please remember to choose, using the manuscript system, whether or not you want to participate in transparent peer review.**

Please note: we allow redactions to authors' rebuttal and reviewer comments in the interest of confidentiality. If you are concerned about the release of confidential data, please let us know specifically what information you would like to have removed. Please note that we cannot incorporate redactions for any other reasons. Reviewer names will be published in the peer review files if the reviewer signed the comments

to authors, or if reviewers explicitly agree to release their name. For more information, please refer to our FAQ page.

Thank you again for your interest in Nature Computational Science. Please do not hesitate to contact me if you have any questions.

Best,
Fernando

--

Fernando Chirigati, PhD
Chief Editor, Nature Computational Science
Nature Portfolio

ORCID

Reviewer #1 (Remarks to the Author):

I appreciate very much the improvements in particular the test of the model verification with the contact law of Young Dupré.

Some (minor) remarks

- Please check the value of the omega (adhesion strength) in FIG S4 as compared to Table 2 as they differ multiple orders of magnitude.

- I understand the behavior of the cells in Fig4 better now. However, with regard to the mesh deformation, the authors write in the rebuttal letter that the term "unphysiological" behavior corresponding to Fig4c left, right was mentioned in the manuscript; however I could not find this in manuscript. Perhaps the behavior explained in the letter could be transferred (or parts of it) to the manuscript.

- I remain a bit critical about how the computational efficiency of the code is communicated in the paper (Fig1e). I have little reason to doubt that their code is efficient compared to other codes, yet the example simulation of a spheroid up to 100000 cells in a day from one cell is a bit misleading because of the unphysiologically large growth rates to achieve this. As the authors mention themselves, this creates an out of equilibrium system cannot be realistic. In my opinion, code efficiency should be tested by comparing the computational time needed to complete T timesteps for various cells numbers (or nodes), or to compare the time needed to simulate production N cells (with physiological growth parameter, i.e. $t_{div} \sim 24h$) with that of the real system. This would allow the user to have a clear view on how long a simulation will take.

- Please indicate whether the code will be able to be run in Windows or Mac (I assume

that it is only Linux , as stated in the code files) ?

- My impression is that the documentation of the code for now is rather limited; in addition working with xml files might be an obstacle for non expert users

Reviewer #1 (Remarks on code availability):

I have been able to download the code but I haven't been able to install or test it because of lack of time and I am currently only working with a Windows systems (which seems not to be supported). I have given the code to a student to test the efficiency and user friendliness it but so far I have no more information.

remarks : in the readme, I was not able to download information from the provided link : https://github.com/SteveRunser/SimuCell3D_v2.git

Reviewer #3 (Remarks to the Author):

I thank the authors for responding to all my comments. I am satisfied with the responses and believe the manuscript is acceptable for publication.

I do recommend some changes to make the tool more useable see code availability box.

Reviewer #3 (Remarks on code availability):

I would recommend providing a docker container or similar to provide easy access to the code and tool. As it stands a potential user requires a linux machine to use (or they need to use a VM as was the case for this reviewer). In addition for a tool of this size I would expect a wiki or similar documentation along with the mentioned git repository. This may be in place for post publication but all I can assess is the downloaded zip which is not sufficient support.

Author Rebuttal, first revision:

Reviewer #1 (Remarks to the Author):

I appreciate very much the improvements in particular the test of the model verification with the contact law of Young Dupré.

Some (minor) remarks

Please check the value of the omega (adhesion strength) in FIG S4 as compared to Table 2 as they differ multiple orders of magnitude.

Indeed, the value was incorrect. Thank you very much for catching that oversight.

I understand the behavior of the cells in Fig4 better now. However, with regard to the mesh deformation, the authors write in the rebuttal letter that the term “unphysiological” behavior corresponding to Fig4c left, right was mentioned in the manuscript; however I could not find this in manuscript. Perhaps the behavior explained in the letter could be transferred (or parts of it) to the manuscript.

As requested, we have added the term “unphysiological” in the sentence describing the morphological regimes.

I remain a bit critical about how the computational efficiency of the code is communicated in the paper (Fig1e). I have little reason to doubt that their code is efficient compared to other codes, yet the example simulation of a spheroid up to 100000 cells in a day from one cell is a bit misleading because of the unphysiologically large growth rates to achieve this. As the authors mention themselves, this creates an out of equilibrium system cannot be realistic. In my opinion, code efficiency should be tested by comparing the computational time needed to complete T timesteps for various cells numbers (or nodes), or to compare the time needed to simulate production N cells (with physiological growth parameter, i.e. $t_{div} \sim 24h$) with that of the real system. This would allow the user to have a clear view on how long a simulation will take.

We have simulated the exponential growth of a tissue from 1 to 500 cells, and measured the average computation time per iteration per number of nodes as a function of the number of threads. These new results are summarized in Fig S6c. A user can now estimate the computational time needed to simulate a certain number of iterations with a specific geometrical resolution.

Please indicate whether the code will be able to be run in Windows or Mac (I assume that it is only Linux , as stated in the code files) ?

The code was indeed developed to run on Linux. However, Windows and Mac users can now also run the program through a Docker container. Windows users have the additional possibility to run the code via a Windows Linux Subsystem. We added these explanations to the README file of our program.

My impression is that the documentation of the code for now is rather limited; in addition working with xml files might be an obstacle for non expert users

Our current documentation provides detailed instructions on installing and operating our program. Additionally, we have written documentation pages covering the following topics:

- Configuration of the simulations
- The structure of the input files required by, and output files produced by our program
- The generation of new initial geometries
- Parameter screens on cluster computers
- Running our program and analyzing its results directly from Python

We will of course value the feedback received from our users and remain committed to enhancing our documentation as necessary to better meet their needs. We consider the use of an XML file for parameterizing simulations as a widely accepted and straightforward method. The advantages of XML over other formats include:

- XML reading and writing software is widely available (the xml Python package, numerous xml parsers in C++, etc.)
- as a structured text format, XML is portable and easily human-readable
- a simple text editor suffices to view or change parameters

We believe that XML is the best possible option that will be most accessible by the majority of users.

Reviewer #1 (Remarks on code availability):

I have been able to download the code but I haven't been able to install or test it because of lack of time and I am currently only working with a Windows systems (which seems not to be supported). I have given the code to a student to test the efficiency and user friendliness it but so far I have no more information.

remarks : in the readme, I was not able to download information from the provided link :

https://github.com/SteveRunser/SimuCell3D_v2.git

We will transfer the code to an open repository and update this link before the final publication of our article.

Reviewer #3 (Remarks to the Author):

I thank the authors for responding to all my comments. I am satisfied with the responses and believe the manuscript is acceptable for publication.

I do recommend some changes to make the tool more useable see code availability box.

Reviewer #3 (Remarks on code availability):

I would recommend providing a docker container or similar to provide easy access to the code and tool. As it stands a potential user requires a linux machine to use (or they need to use a VM as was the case for this reviewer). In addition for a tool of this size I would expect a wiki or similar documentation along with the mentioned git repository. This may be in place for post publication but all I can assess is the downloaded zip which is not sufficient support.

As recommended, we have implemented the option to install and run our program via a Docker container. Additionally, we have updated the README file in our repository to provide clear instructions for this installation process.

We have added pages explaining the following topics to the repository documentation:

- Configuration of the simulations
- The structure of the input files required, and output files produced by our program
- The generation of new initial geometries
- Parameter screens on cluster computers
- Running our program and analyzing its results directly from Python

Final Decision Letter:

Date: 8th March 24 10:22:54

Last Sent: 8th March 24 10:22:54

Triggered By: Kaitlin McCardle

From: kaitlin.mccardle@us.nature.com

To: dagmar.iber@bsse.ethz.ch

BCC: fernando.chirigati@us.nature.com,rjsproduction@springernature.com,computationalscience@nature.com,kaitlin.mccardle@us.nature.com

Subject: Decision on Nature Computational Science manuscript NATCOMPUTSCI-23-0381C

Message Dear Professor Iber,

:

We are pleased to inform you that your Resource "SimuCell3D: 3D Simulation of Tissue Mechanics with Cell Polarization" has now been accepted for publication in Nature Computational Science.

Once your manuscript is typeset, you will receive an email with a link to choose the appropriate publishing options for your paper and our Author Services team will be in touch regarding any additional information that may be required.

Please note that *Nature Computational Science* is a Transformative Journal (TJ). Authors may publish their research with us through the traditional subscription access route or make their paper immediately open access through payment of an article-processing charge (APC). Authors will not be required to make a final decision about access to their article until it has been accepted. Find out more about Transformative Journals

Acceptance of your manuscript is conditional on all authors' agreement with our publication policies (see <https://www.nature.com/natcomputsci/for-authors>). In particular your manuscript must not be published elsewhere and there must be no announcement of the work to any media outlet until the publication date (the day on which it is uploaded onto our web site).

Before your manuscript is typeset, we will edit the text to ensure it is intelligible to our

wide readership and conforms to house style. We look particularly carefully at the titles of all papers to ensure that they are relatively brief and understandable.

Once your manuscript is typeset, you will receive a link to your electronic proof via email with a request to make any corrections within 48 hours. If, when you receive your proof, you cannot meet this deadline, please inform us at rjsproduction@springernature.com immediately.

If you have queries at any point during the production process then please contact the production team at rjsproduction@springernature.com.

You may wish to make your media relations office aware of your accepted publication, in case they consider it appropriate to organize some internal or external publicity. Once your paper has been scheduled you will receive an email confirming the publication details. This is normally 3-4 working days in advance of publication. If you need additional notice of the date and time of publication, please let the production team know when you receive the proof of your article to ensure there is sufficient time to coordinate. Further information on our embargo policies can be found here:

<https://www.nature.com/authors/policies/embargo.html>

We welcome the submission of potential cover material (including a short caption of around 40 words) related to your manuscript; suggestions should be sent to Nature Computational Science as electronic files (the image should be 300 dpi at 210 x 297 mm in either TIFF or JPEG format). We also welcome suggestions for the Hero Image, which appears at the top of our home page; these should be 72 dpi at 1400 x 400 pixels in JPEG format. Please note that such pictures should be selected more for their aesthetic appeal than for their scientific content, and that colour images work better than black and white or grayscale images. Please do not try to design a cover with the Nature Computational Science logo etc., and please do not submit composites of images related to your work. I am sure you will understand that we cannot make any promise as to whether any of your suggestions might be selected for the cover of the journal.

Best regards,

--

Kaitlin McCardle, PhD
Senior Editor, Nature Computational Science
Nature Portfolio

P.S. Click on the following link if you would like to recommend Nature Computational Science to your librarian: <https://www.springernature.com/gp/librarians/recommend-to-your-library>

** Visit the Springer Nature Editorial and Publishing website at www.springernature.com/editorial-and-publishing-jobs for more information about our career opportunities. If you have any questions please click here.**